



# 3D cloud envelope and cloud development velocity from simulated CLOUD/C3IEL stereo images

Paolo Dandini[1], Céline Cornet[1], Renaud Binet[2], Laetitia Fenouil[2], Vadim Holodovsky[3], Yoav Y. Schechner[3], Didier Ricard[4], and Daniel Rosenfeld[5]

[1]Laboratoire d'Optique Atmosphérique, CNRS/Université de Lille, Villeneuve d'Ascq, FR
[2]Centre National d'études spatiales, Toulouse, FR
[3]Viterbi Faculty of Electrical and Computer Engineering, Technion - Israel Institute of Technology, Haifa 32000, Israel
[4]CNRM, Université de Toulouse, Météo-France, CNRS, Toulouse, France
[5]Institute of Earth Sciences, The Hebrew University of Jerusalem, Jerusalem, Israel

**Correspondence:** P. Dandini (paolo.dandini@univ-lille.fr, dandini.paolo96@gmail.com)

**Abstract.** A method to derive the 3D cloud envelope and the cloud development velocity from high spatial and temporal resolution satellite imagery is presented. The CLOUD instrument of the recently proposed C3IEL mission lends itself well to observing at high spatial and temporal resolutions the development of convective cells. Space-borne visible cameras simultaneously image, under multiple view angles, the same surface domain, every 20 s over a time interval of 200 s. In this paper, we

present a method for retrieving cloud development velocity from simulated multi-angular-high-resolution TOA radiance cloud fields. The latter are obtained via the radiative transfer model 3DMCPOL, for a deep convective cloud case generated via the atmospheric research model Meso-NH, and via the image renderer Mitsuba for a cumulus case generated via the atmospheric research model SAM. Matching cloud features are found between simulations via block matching. Image coordinates of tie points are mapped to spatial coordinates via 3D stereo reconstruction of the external cloud envelope for each acquisition. The

accuracy of the retrieval of cloud topography is quantified in terms of RMSE and bias that are respectively, less than 25 m and 15 m for the horizontal components and less than 40 m and 25 m for the vertical component. The inter-acquisition 3D velocity is then derived for each pair of tie points separated by 20 s. An independent method based on optimizing the superposition of the cloud top, issued from the atmospheric research model, allows to obtain a ground estimate for the velocity from two consecutive acquisitions. The distribution of retrieved velocity and ground estimate exhibits small biases but significant dis-

crepancy in terms of distribution width. Furthermore, the average velocities derived from the mean altitude from ground for a cluster of localized cloud features identified over several acquisitions, both in the simulated images and in the physical point cloud, are in good agreement.



## 1 Introduction

Large uncertainties concerning the evolution of climate still prevail (IPCC, 2021). Based on the current knowledge, these uncertainties are for a large part ascribable to the interactions of clouds with aerosols and to the role played by clouds in the general circulation. In this respect, aerosols-clouds interaction is one of the fundamental subjects of the 2021 IPCC report (chapter 8) and the link between clouds, general circulation and climate sensitivity, of major importance (Bony et al. 2015), is one of the seven top challenges of the World Climate Research Program (WCRP). Despite the progress made over the last thirty

years, clouds still represent a large source of uncertainty for meteorological models at any scale, from Large Eddy Simulations (LES) to Numerical Weather Predictions (NWPs) and in turn for General Circulation Models (GCMs). Our understanding and ability to constrain cloud processes have to improve if a better representation of clouds in the models is to be achieved. The current discrepancies between observations and climate simulations are mainly associated to the still relatively little knowledge of the physical processes from which they originate and develop. In particular, current observations do not allow to resolve the

turbulent eddies that drive cloud development. These are the result of the entrainment of air in the cumulus clouds (Sherwood et al., 2014; Donner et al., 2016) deriving from the combination of air streams ascending and descending within the cloud. Although estimates of convective buoyancy and entrainment rate (Luo et al., 2010) were obtained from the A-Train data while CloudSat observations were used to observe the relationship among convective intensity, entrainment rate, convective core width, and outflow height (Takahashi et al., 2017), these relations remain highly challenging to reproduce in the models. The

cloud development velocity and the characterization of the turbulent structures associated with it, resulting from the air streams inside the clouds, are key for understanding cloud precipitation systems and for determining the interaction between cumulus ensembles and large-scale atmospheric environment (Hamada and Takayabu, 2016). Unfortunately, given the intrinsic difficulty associated to cloud observations only a limited amount of direct measurements exists, such as: in situ measurements of vertical wind in oceanic convective cumulus clouds (Lucas et al., 1994), cumulonimbus vertical velocities from wind profilers

(LeMone and Zipser, 1980), vertical velocities in the convection from Doppler radar measurements (Heymsfield et al., 2010), vertical air motion from ground based and air borne doppler radars (Collis et al., 2013), velocity retrievals from ground based wind profiling radar (Giangrande et al., 2013), profiler observations to provide vertical velocity statistics on the full spectrum of tropical convective clouds (Schumacher et al., 2015). All these works have provided reliable information but they do not allow resolving small cloud structures and lack in terms of spatio-temporal coverage. Satellite observations have been exploited to

overcome such limitations. Following the pioneering study by Adler and Fenn (1979) and the recent work of Luo et al. (2014), Hamada and Takayabu (2016) estimate convective cloud top vertical velocity from the decreasing rate of infrared brightness temperature observed by the Multi-functional Transport Satellite-1R (MTSAT-1R). However, when it comes to IR cloud top retrieval, complicated thermodynamics can cause problems for IR height assignments and the low resolution leads to underestimating the cloud top height. Stereo imaging methods, on the other hand, have become more and more utilized with the new

generation of GEO satellites. The wide field of view and continuous measurements obtainable from geostationary satellites allows a considerable number of samples independent of region and season. Stereo methods allow direct observation of Atmospheric Motion Vectors (AMV) by tracing cloud or water vapour features over multiple synchronous images. In a feasibility





study (Horvath and Davis, 2000) cloud motion vectors were derived from simulated multi-angle images to be obtained from the Multiangle Imaging Spectroradiometer (MISR); horizontal wind vectors (advection) and cloud heights were retrieved over

a mesoscale domain of about 70 km x 70 km. The first retrievals from actual data (Horvath and Davis, 2001) were consistent with the prelaunch error estimates of $\pm 3$ m/s and $\pm 400$ m for winds and heights, respectively. These retrievals were obtained for the first time from the polar orbiting spacecraft Terra. The main limitations of their method is the fact that vertical cloud motion is neglected and a constant horizontal cloud advection over the domain is assumed which under intense convection, for instance, may lead to unreliable retrieved winds. Similarly cloud top heights and winds, retrieved from the Meteosat geo-

stationary satellites (Seiz et al., 2007) via stereo imaging, were tested against retrievals from MISR and MODIS observations. One of the main drawbacks in this case is the time difference between Meteosat satellites which significantly affect matching accuracy. The multi-spectral Stereo Atmospheric Remote Sensing (STARS) employs stereoscopic imaging technique on two satellites (CubeSats) to simultaneously retrieve cloud motion vectors (CMVs), cloud-top temperatures (CTTs), and cloud geometric heights (CGH) from multi-angle, multi-spectral observations of cloud features (Kelly et al., 2018). In another work Seiz

and Davies (2006) present a feasibility study for the 3D reconstruction of cloud geometry, at a resolution of 1 km, from MISR data and via stereo imaging.

    Cloud structure was also recovered from simulated and empirical airborne multi-view images (Levis et al., 2015), emulating the JPL's Airborne Multiangle SpectroPolarimetric Imager (AirMSPI) (Diner et al. 2013). This was done by computed tomography (CT), which inverts a 3D radiative transfer model, to retrieve the extinction coefficient in 3D. 3D cloud geometry is an

important product for constraining the retrieval of cloud properties.

    Loeub et al. (2013) attained 3D volumetric cloud retrieval from simulated multiple views through stochastic scattering tomography. Ronen et al. (2021) derived spatio-temporal 4D cloud CT reconstruction, using space-borne simulations and AirMSPI data. 4D CT generalizes 3D CT, enabling the use of a few imaging platforms, which move (orbit) while capturing multiangular data. Sde-Chen et al. (2021) devised a neural network for spaceborne 3D cloud CT, leading to 5-order of

magnitude acceleration, relative to Levis et al., 2015. While these studies focused on multi-view cloud imaging from above, Veikherman et al. (2014) and Aides et al. (2020) performed 3D recovery of cloud geometry using a ground-based distributed imaging system, which consists of a wireless network of sky cameras.

    We draw on these previously published works on stereo retrieval of cloud geometry and cloud motion vectors and present a method for the 3D retrieval of cloud envelope and cloud development velocity at high spatial resolution from multi-angular,

multi-view imagery. This work was carried out during phase A in preparation for the C3IEL (Cluster for Cloud Evolution, Climate and Lightning) space mission. Although at present we are not aware of when C3IEL will make it to the next phase of the project, we were able to follow through on this work that relies entirely on realistic image simulations. C3IEL will consist in a train of two to three satellites on a sun-synchronous orbit observing simultaneously the same cloud scene every 20 s over 200 s. The CLOUD cameras of the C3IEL mission will image the same cloud domain at unprecedented high spatial

and temporal resolutions. While no observations are yet available, image simulations were used to present the method. Results from this work are expected to shed light on whether the use of a third satellite leads to significant improvement in terms of retrieved products. Simulations were obtained for two test cases, a deep convective cloud case and a trade wind cumulus case.





This work is organized as follows. In section 2, the principle of measurement and the methodology used to develop and assess the retrieval algorithm is presented. In section 3, the two cloud cases are presented together with the physical model from which they are derived and the radiance simulators used. Also the reference cloud envelope is defined. In section 4 we present the method to obtain 3D cloud reconstruction and how we test the 3D retrieval against the reference cloud. Section 5 is devoted to the method for the retrieval of the stereo cloud development velocity and the method for deriving a ground estimate of cloud development velocity; in the same section results and discussion of the testing of the 3D velocity retrieval are also presented. In section 6 conclusions are drawn.

## 2 Principle of measurement and methodology

### 2.1 C3IEL mission and the CLOUD cameras

C3IEL (Cluster for Cloud Evolution, ClimatE and Lightning) is a project for a joint space mission between the French (CNES) and the Israeli (ISA) space agencies. It relies on a cluster of synchronized nano-satellites mainly focused on the study of convective clouds at high spatial resolution for the retrieval of cloud updraft and the monitoring of water vapor and lightning activity. The different nano-satellites of the C3IEL mission (2 or 3 to be defined) will carry visible spectrum cameras (CLOUD) measuring at a spatial resolution of about 20 meters, near-infrared imagers (WV) measuring in and around water vapor absorption bands (500 m resolution), optical lightning sensors and photometers (LOIP). The observational strategy for the imagers will consist in multi-angular observations of a given cloud scene during 200 s with instantaneous stereoscopic pairs or triplets captured every 20 s (11 acquisitions A1-A11 see Fig. 1) corresponding to the life time of cloud perturbations at small scale. Lightning observations will be made continuously during the same time. The measurements of these space-borne sensors will consequently simultaneously document the vertical cloud development retrieved by a stereoscopic method, the lightning activity and the distribution of water vapor at a high resolution by exploiting the multi-angle acquisitions.

The CLOUD cameras, which is the focus of this work, will simultaneously observe the same pixel area under different view angles. Figure 2 shows simulations of some of the successive CLOUD observations corresponding to acquisitions A1 (far from nadir), A4 and A7 (close to nadir). The distance between satellites, referred to as baseline B, will be about 150 km.



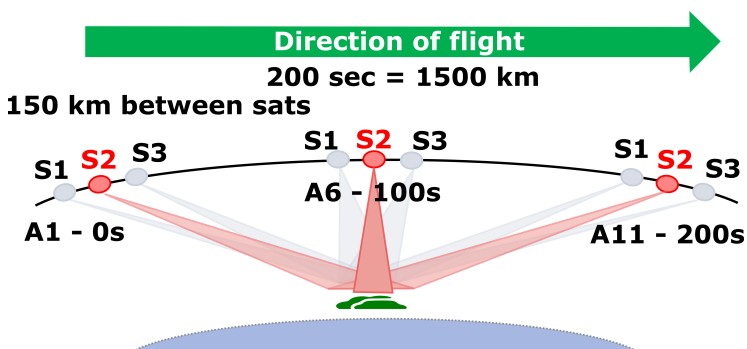

**Figure 1.** CLOUD/C3IEL observational strategy: multi-view and simultaneous imaging from satellites flying at an altitude of 600 km. Acquisitions A1, A6 and A11 occur at time t=0 s, t=100 s and t=200 s respectively.



(a) Sat 1 A1 - $\theta_v = 59.1°$  (b) Sat 2 A1 - $\theta_v = 53.6°$  (c) Sat 3 A1, - $\theta_v = 46.8°$

(d) Sat 1 A4, - $\theta_v = 38.3°$  (e) Sat 2 A4, - $\theta_v = 27.9°$  (f) Sat 3 A4, - $\theta_v = 15.6°$

(g) Sat 1 A7, - $\theta_v = 1.7°$  (h) Sat 2 A7, - $\theta_v = 12.3°$  (i) Sat 3 A7, - $\theta_v = 25.1°$

**Figure 2.** Realistic CLOUD/C3IEL radiance simulations of a deep convective cloud corresponding to A1 (Fig. 2a, 2b, 2c), A4 (Fig. 2d, 2e, 2f) and A7 (Fig. 2g, 2h, 2i), $\theta_v$ is the angular distance of the camera center from Nadir. The sun, at a incidence angle equal to $\theta_s = 13.6°$, illuminates from the right hand side of the images.



The CLOUD cameras will image a footprint area of about 80 km x 45 km (Notice that Fig. 2 shows an area of 15km x 15km) with the aim to obtain an accuracy of the order of a few m/s in terms of vertical development speed. The operative wavelength will be 670 nm where cloud contrast is larger, molecular and aerosol effects, as well as surface brightness being less important. It should be noticed that satellite is moving from North to South and while the across track resolution remains almost constant the along track resolution is increasing for tilted views. This leads to an increase of the footprint in the along track direction and a reduction of the ground sampling distance (GSD). Finally, there are image artifacts, mostly visible in the close-to-Nadir views (A4, A7), which are associated to the cyclic replication of the cloud field via the Monte-Carlo code. Most of these artifacts are excluded from the calculations presented in this work, specifically when performing stereo processing. This is achieved by opportunely setting the boundaries of the region of interest (ROI) of the images to a replicated domain.

### 2.1.1 The end to end simulator: methodology

In order to carry out this work, while no real C3IEL data yet exist, we had to rely on simulations. A flow diagram of the methodology used (the end-to-end CLOUD simulator) for the retrieval of cloud development velocity is given in Fig. 3. The physical parameters (from the MESO-NH model for the deep convective case and from The System for Atmospheric Modeling (SAM) for the cumulus case), were converted into optical properties which are then fed to the radiative transfer model, 3DMCPOL (Cornet et al. 2010) for the deep convective case and to the image renderer Mitsuba for the cumulus case (Wenzel, 2010), for radiance rendering. Simulations were obtained for perspective projection cameras. The geometry of the external cloud envelope (stereo (ST) cloud) is retrieved for each acquisition from image pairs or triplets and tested against the GT cloud envelope, to which we will refer to as Ground Truth (GT) cloud envelope, obtained from the cloud physical properties (extinction coefficient or liquid water content) input to the radiance renderers. From these synthetic observations, corresponding cloud pixels, matching cloud features that we refer to as tie points, were tracked between acquisitions, then 3D reconstruction of the cloud envelope was used to map image coordinates to space coordinates for all tie points. In this way the between-acquisition mean cloud velocity vector (ST velocity vector) was obtained by dividing the distance traveled by the inter acquisition time (20 s). The velocity retrieval was done only for the deep convective case and was, analogously to the geometry retrieval, tested against a Ground Estimation of cloud velocity (GE velocity vector). The latter was derived from the GT cloud envelopes and independently from the stereo retrieval.





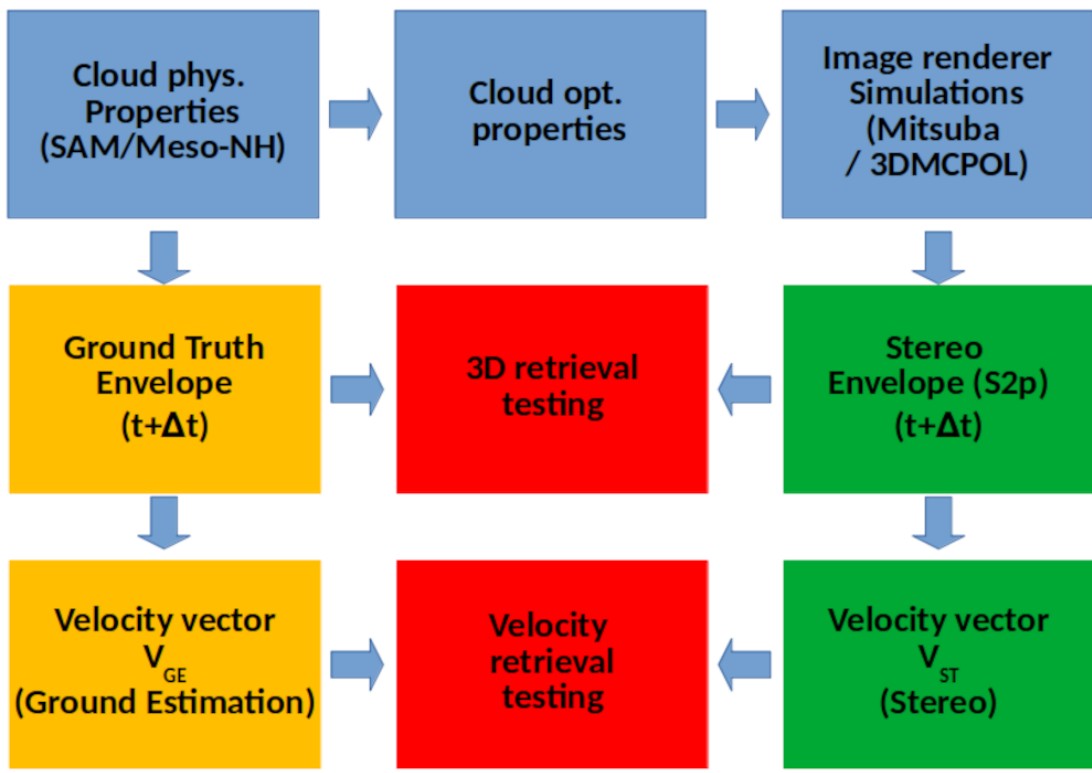

**Figure 3.** End-to-end CLOUD/C3IEL simulator: methods and testing.



## 3 Data: the CLOUD simulations

### 3.1 Test case 1 - A deep convective cloud

#### 3.1.1 Physical properties from LES

The first test case is a deep convective cloud simulated with the non-hydrostatic mesoscale atmospheric model Meso-NH
model (Lac et al., 2018), used in LES mode. Fig. 4 shows the vertical section, at 22.5 km, of total water content, as well as
vertical and horizontal wind components. Simulations were done (Strauss et al., 2019) with a 3D turbulence scheme (Cuxart
et al., 2000), the microphysical scheme ICE3 including five hydrometeor species (Pinty and Jabouille, 1998) and no radiation
scheme. Cloud model domain, resolution and assumptions are given in table 1. The spatial resolution of the domain is 50 m. As
the entire cloud scene (1600x1600x260 cells) is too large to be handled by 3DMCPOL, sub-domains of 300x300x160 cloud
cells were extracted from the initial scene. With a horizontal extent of around 10 km and a cloud top at about 9 km, this is
a case of well-developed convective cloud characterized by relatively strong in-cloud air stream speeds, of up to 20 m/s, for
both horizontal and vertical components. Consistently, relatively large values of water vapor mixing ratio (not shown), up to
$16 \ 10^{-3}$ kg/kg, are observed up to 2 km from ground.

| Meso-NH | |
|---|---|
| Initial Domain | 80 km x 80 km x 20 km |
| Resolution | dx = dy = 50 m; dz = 50 m from z = 0 to 13000 m |
| Time step | 20 s |
| Assumptions: | Idealized simulation : <br><br> 1. Atmospheric instability : humidity and temperature profile from Weisman and Klemp (1984) <br><br> 2. Small temperature perturbations (white noise) in the lowest layers (below 1000 m) <br><br> 3. Moderated wind shear <br><br> 4. Prescribed latent and sensible heat at the surface (200 and 350 W/m$^2$) <br><br> 5. Surface roughness length (0.035 m) <br><br> 6. No Coriolis force |

**Table 1.** Settings of Meso-NH for the simulations of the deep convective cloud.

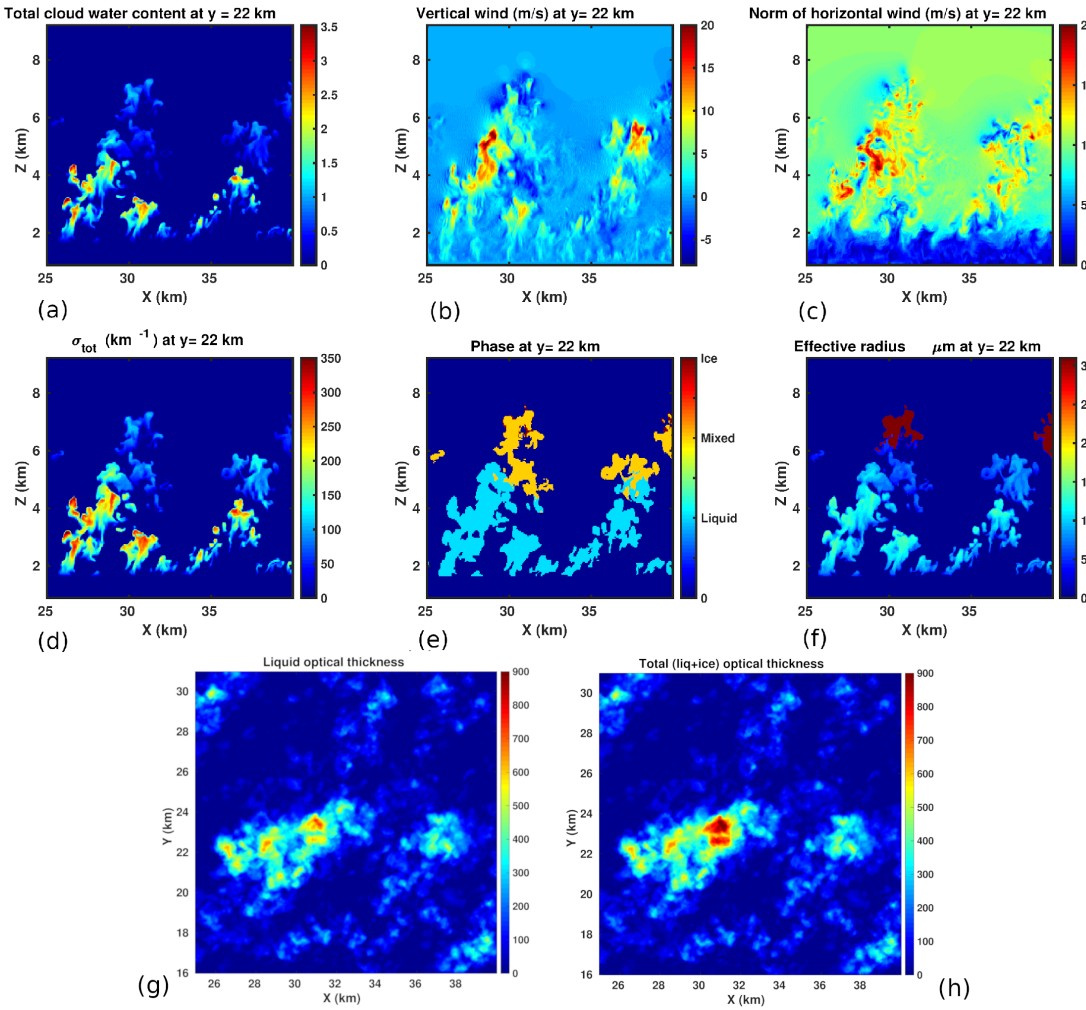

**Figure 4.** Deep convective cumulus physical properties. Fig. 4a, 4b, 4c: vertical section (y=22.5 km) of cloud total water content, vertical and horizontal wind components, respectively - Fig. 4d, 4e, 4f: vertical section (y = 22.5 km) of total extinction coefficient, cloud phase and effective radius, respectively. Value of 31 (dark red) is associated to voxels where the mean ice phase function is used - Fig. 4g, 4h: liquid optical thickness and total optical thickness (liquid + ice), respectively.





### 3.1.2 From physical to optical properties

Cloud optical properties such as extinction coefficient, single scattering albedo and phase function (PF) have to be fed into the 3DMCPOL in order to compute cloud radiances. For the liquid phase, they are quantified from the water content and effective radius, using the Mie theory of scattering. As the version of the model Meso-NH model used here does not compute the particle size distribution (PSD), we find the effective radius of the water droplets by using Eq. 1 (Martin et al., 1994):

$$r_{eff} = 1000 \left( \frac{3r_l}{4\pi\rho k N} \right)^{\frac{1}{3}} \tag{1}$$

where $r_l$ is the mass of liquid water per unit volume of air, $\rho$ is density of water, k is a constant, 0.8 or 0.67 for maritime or continental scenario, respectively. The droplet number per unit volume N is set to 300 $cm^{-3}$. Eq. 1 was established for stratocumulus clouds and although not completely appropriate for deep convective clouds, it reproduces fairly well the observed values (Freud and Rosenfeld, 2012). Furthermore, this is not going to affect in any way the assessment of the stereo retrieval. The liquid extinction coefficient, $\sigma_{liq}(km^{-1})$, is calculated using Eq. 2:

$$\sigma_{liq} = \frac{3r_l 10^{-3}}{2(\rho r_{eff} 10^{-6})1000} \tag{2}$$


For the ice part, the Baran's parameterization (Baran et al., 2014) is used to compute extinction coefficient, ice single scattering albedo $\omega_{ice}$ and PF from the ice water content and ice temperature (In this case, the extinction coefficient is obtained from look up tables of absorption and scattering coefficients). A modified Henyey-Greenstein phase function (Baran et al. 2001) is then computed from the asymmetry parameter g and the average ice phase function is taken. For the mixed phase, 165   the extinction coefficient is the sum of the liquid and ice extinction coefficient while the single scattering albedo $\omega_{mix}$ is taken as the average of $\omega_{ice}$ and $\omega_{liq}$ weighted by the respective extinction coefficient. The cloud phase associated to the larger extinction coefficient (ice vs liquid) is selected. As a result, the vertical section of total extinction $\sigma_{tot}$, cloud phase and $r_{eff}$ are plotted in Fig. 4d, Fig. 4e, Fig. 4f. Relatively large values of $\sigma_{tot}$ up to 300 km$^{-1}$ are observed between 3 and 4 km where mostly liquid particles are found; as well as significantly lower values of $\sigma_{tot}$, above 4.5 km, where transition from liquid to 170   mixed phase occurs; also noticeably, the position of the peak of the particle size distribution varies with altitude, with larger ice crystals above 6 km and smaller cloud particles below. Liquid and total optical thickness $\tau_{liq}$ and $\tau_{tot}$ (see Fig. 4g and Fig. 4h), obtained by integrating $\sigma_{liq}$ and $\sigma_{tot}$ respectively, over the geometric extent of the cloud, have also been plotted. Noticeable are the very high cloud optical thickness (COT) values of up to 700 typical of this type of cloud.



### 3.1.3   The radiative transfer - 3DMCPOL

Shadowing and illumination effects are of great importance when it comes to detecting identical cloud features from pairs or triplets of CLOUD images. That is why clouds cannot be assumed to be plane-parallel and homogeneous when computing radiances. Moreover, the high resolution of the CLOUD cameras makes the independent pixel approximation no longer valid. In order to account for these effects, a radiative transfer model allowing calculations in a 3D atmosphere is necessary. With the aim to obtain realistic observations of the CLOUD sequence of acquisitions, we use the forward 3D Monte-Carlo radiative
transfer model (Cornet et al., 2010). 3DMCPOL was initially thought for computing radiances in orthographic mode that is having parallel output directions for the radiance of each camera pixel. As the satellite can be considered to be at infinity this assumption is valid for the simulation of a small part of a large swath or when 3D radiative transfer is used to simulate sub-pixel heterogeneity (Cornet et al., 2018). However, to render a large portion of the images at a high spatial resolution in camera mode, the orthographic assumption is no more valid. Furthermore, pixel size increases with the distance from the image center and so does the field of view for tilted views. To account for this particular geometry, 3DMCPOL uses the output of an orbit,
attitude and camera simulator developed by CNES and dedicated to the C3IEL mission. This simulator is coupled with CNES geometric library (Euclidium) in order to obtain grids at different altitudes giving the correspondence between a 3D position (x,y,z) in the medium to the index (i,j) of the camera image and to the output direction defined by a zenithal and azimuthal angles. From these grids, trilinear interpolation gives access to the image pixel line and column and to the line of sight to apply
local estimate method used in 3DMCPOL. Sampling of the grids is managed so that the trilinear interpolation does not bias the samples within 0.1 pixel accuracy. In this way, the sphericity of the orbit and the orientation of the satellites are accounted for. Another advantage of the simulator is the possibility to include image distortion. Due to computational limitations, we were not able to compute the entire field of view of the cameras, which is 80 km x 45 km composed of 4608 by 2592 pixels with a size of 17 m at Nadir. Consequently, the input cloud was chosen to be a 3D medium with 300x300x200 voxels with 50
m resolution in the three directions, corresponding to an horizontal extent of the cloudy medium of 15 km x 15 km. Figure 2 shows 3DMCPOL simulations for the acquisition A1 ($T_0$), off nadir (top three figures), acquisition A4 ($T_0$+ 60 s) (middle images - closer to nadir) and acquisition A7 ($T_0$+ 120 s) approximately at nadir (bottom figures), respectively. Each simulation is identified by satellite and acquisition numbers ST$N_s$A$N_a$($N_s$=1,2,3 - $N_a$=1,2,3,4,5,6,7,8,9,10,11).

### 3.2   Test case 2 - The trade wind cumulus

#### 3.2.1   Physical and optical properties

In this section, we turn to the cumulus cloud case generated via the System for Atmospheric Modeling (SAM), a non-hydrostatic anelastic model that simulates cloudy atmospheres in a wide range of scales, from boundary-layer turbulence to hurricanes. It can be configured as a LES model to investigate cloudy or cloud-free boundary layers, or as a Cloud-Resolving Model (CRM) to study deep convective clouds and meso-scale cloud systems. The SAM has been successfully ported to many
different computing platforms including massively parallel supercomputers. The original version of the model is discussed by Khairoutdinov and Randall (Khairoutdinov and Randall, 2003). The CRM that represents subgrid processes in the CSU (Col-





orado State University) multi-scale modeling framework (MMF) has a relatively simple representation of cloud microphysics. This scheme is fast, but it does not allow for the explicit representation of freezing/melting of hydrometeors, size sorting of falling precipitation and aerosol effects on clouds (also known as aerosol indirect effects). Phases of condensed water are diag-

nosed from the temperature. The physical parameters obtained via SAM for the trade wind cumulus fields are the Liquid Water Path (LWP), the particle number concentration (NC), the wind velocities, the particle effective radius and the Liquid Water Content (LWC). The cloud field domain consists of 512x512x200 voxels each with size 20x20x20 $\text{m}^3$. The vertical section of water content (see Fig. 5a) and the relatively low winds (see Fig. 5b and Fig. 5c), of up to 10 m/s, were plotted for a horizontal resolution of 20 m. With a cloud base between 1 and 2 km, this test case is an example of not well-developed cumulus cloud

as often found at mid-latitudes. For the liquid phase, which is the only cloud phase taken into account in this case, the optical properties were quantified from the liquid water content and the effective radius using the Mie theory of scattering. Extinction coefficient $\sigma_{liq}$ and the phase function (PF) were then fed into the rendering code Mitsuba (Wenzel, 2010) to obtain the cloud radiance. In comparison to the deep convective case we notice vertical winds and COT values (see Fig. 5g) up to four and seven times lower, respectively.







**Figure 5.** Trade wind cumulus physical properties. Fig. 5(a), Fig. 5(b), Fig. 5(c): vertical section for y=7 km, from left: liquid water content, vertical and horizontal wind components, respectively - Fig. 5(d), Fig. 5(e), Fig. 5(f): vertical section (y=7 km), from left: liquid extinction coefficient, cloud phase and effective radius. - Fig. 5(g): Cloud liquid optical thickness.



### 3.2.2 The rendering code - Mitsuba

The Mitsuba rendering code enables simulation of radiance as observed from a perspective camera. Similarly to 3DMCPOL, Mitsuba implements 3D Monte-Carlo calculations. However, Mitsuba back-propagates radiance from each camera to the sun. Mitsuba is a relatively fast rendering software. However, it is not based on physical first-principles, that can be derived for an atmospheric medium. This affects the radiometric reliability of the images. However, this effect is not significant in the context of 3D geometric surface reconstruction and tracking of feature points. Here are the assumptions that underlie Mitsuba, and our use of it:

1. Only one particle type exists in the medium: a cloud droplet. Hence, the effects of scattering by molecules and aerosols are not accounted for. This is tolerable in our context, because the system is planned to image clouds around optical wavelength of 670 nm. At this wavelength, the effect of molecules and aerosols on space-borne images is small.

2. The angular distribution of scattering is set by the PF. The PF can be derived from physical first-principles, using Mie theory and the droplet size distribution. The size distribution has some parameters, including the effective radius $r_{eff}$, which vary across the domain. In contrast, Mitsuba assumes a single particle type, a spatially invariant $r_{eff}$ and a Henyey-Greenstein PF. This is a discrepancy from physics. However, this discrepancy is moderated by multiple scattering, which is dominant in clouds. We use a value of 0.85 for the Henyey-Greenstein anisotropy parameter (Mayer, 2009). We use $r_{eff}$ and LWC as provided by the SAM, see section 3.2.1). The liquid extinction coefficient, $\sigma_{liq}(km^{-1})$, is calculated via Eq. 2.

3. We simulate clouds over the ocean in the red spectrum. So, we set the ground albedo to zero.

Simulations were obtained for a constellation of three satellites orbiting at an altitude of about 600 km, separated by a distance of 150 km, and imaging the same ground footprint of about 10.24x10.24 km$^2$. The sun is at 22.5° from zenith and the three satellites carry the same camera with a field of view of 1° (field of view in x and y directions are the same). The domain size along the vertical (z) is 4 km. The sensor size is 500x500 px, for a pixel footprint at the ground of approximately 20x20 m$^2$. Each pixel radiance is simulated using 4096 photons. In order to test the algorithm for the retrieval of cloud geometry, 11 simulations, obtained every 20 s over a time range of 200 s, were used. Some of them, namely those obtained at large distance from nadir (t=0 s, 200 s) and those closer to nadir (t=100 s) are shown in Fig. 6. Satellites are travelling North-South on a linear orbit and the same considerations about across and along track resolutions for tilted views, seen for the deep convective case, still stand. In Fig. 6, notice the bright cloud tops and shaded cloud sides. In the dark-shaded regions, feature identification is more difficult.



(a) Sat 1 - 0s                          (b) Sat 2 - 0s                          (c) Sat 3 - 0s

(d) Sat 1 - 100s                        (e) Sat 2 - 100s                        (f) Sat 3 - 100s

(g) Sat 1 - 200s                        (h) Sat 2 - 200s                        (i) Sat 3 - 200s

**Figure 6.** Realistic CLOUD/C3IEL radiance simulations for a cumulus cloud case and corresponding to acquisitions A1, A6 and A11.





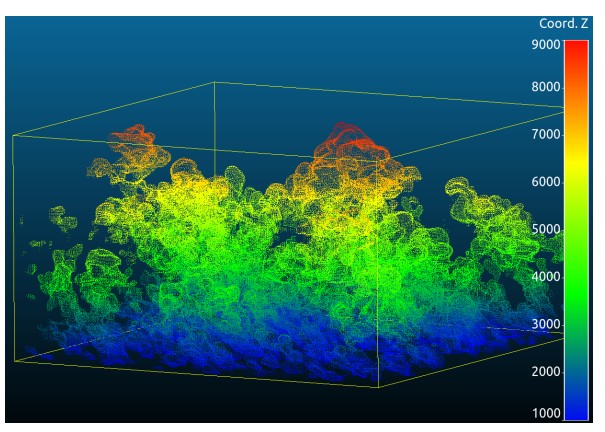
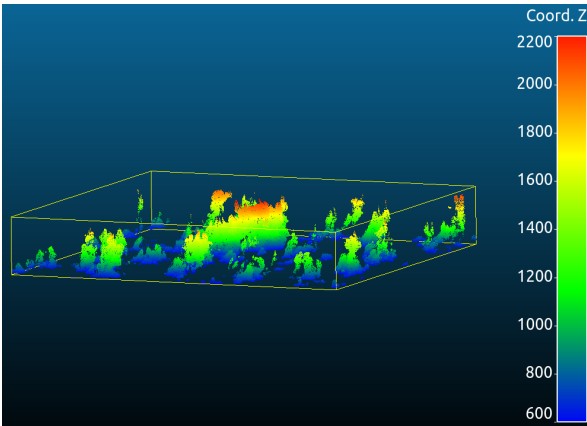

(a) GT point cloud for the deep convective cloud case (A5).  (b) GT point cloud for the trade wind cloud case (A6).

**Figure 7.** 3D cloud envelope referred to as GT (Ground Truth ) point clouds. Point colour shows the Z coordinate (altitude from ground expressed in m).

### 3.3 The ground truth point cloud

To test the stereo algorithm as a function of the geometry and the number of satellites, the ground truth (GT) point cloud was

250 derived from the LWC, issued from the physical models, for both cloud cases. The GT point cloud is defined for each direction x, y and z as the boundary cloud voxels corresponding to the cloud contour and was determined for all cross sections parallel to the xy, yz and xz planes and for each cloud cell. As for testing of the stereo cloud (ST cloud) against the GT cloud, covered in section 4, the GT cloud was mapped from East-North-Up (ENU) to UTM (Universal Transverse Mercator) coordinates. The GT point cloud so obtained is shown in Fig. 7 for the deep convective case and for the trade wind cumulus, Fig. 7a and Fig. 7b

respectively. For simplicity in what follows we will refer to it as the GT cloud or $GT A_N$ ($A_N$=acquisition number) when the acquisition number is specified.





## 4 3D cloud envelope: geometry retrieval

In this section we first discuss the stereo retrieval of cloud geometry, in other words the 3D space retrieval of the cloud pixels
sensed by the cameras, then the method applied for testing the retrieval and eventually the results of the testing, in paragraph
4.2.

### 4.1 The 3D reconstruction: dense matching and triangulation

For each acquisition of synthetic pairs or triplets, retrieval of the cloud envelope (ST cloud), is performed. On general grounds,
for each image point the corresponding scene point depth (i.e. distance from the camera) is determined by first finding matching
pixels (i.e. pixels showing the same scene point) in the other image(s) and then applying triangulation to the found matches to
determine their depth. The retrieval was done by means of the stereo reconstruction algorithm s2p (Satellite Stereo Pipeline)
(De Franchis et al., 2014). The use of this algorithm requires knowledge of the analytical camera models that are an inverse
(see eq. 3 and 4) and direct (see eq. 7 and 8) projection models of the camera mapping from geographical coordinates (Lat,
Long, Height) to image coordinates (r, c) and vice versa, respectively. Such models are expressed as the ratio of polynomials
of degree Nth whose coefficients, the RPCs (Rational Polynomial Coefficients), are fit to the mapping of the physical camera
model for each camera and each orientation. Such calculations were done for polynomials of degree 3 ($p_1$,...,$p_8$) according to
the procedure described by Tao and Hu (Tao and Hu, 2001) for the cumulus case and via the Euclidium library for the deep
convective case.

$$r_n = \frac{p_1\left(lat_n, long_n, height_n\right)}{p_2\left(lat_n, long_n, height_n\right)} \tag{3}$$

$$c_n = \frac{p_3\left(lat_n, long_n, height_n\right)}{p_4\left(lat_n, long_n, height_n\right)} \tag{4}$$

The s2p process pipeline can be summarized as follows: input images are first cut into tiles, where the cameras are assumed
to be affine (i.e. perspective projection). With regard to Fig. 8, the input reference (ref) and secondary images (sec) are first
rectified (rec ref, rec sec) to simplify the search of matching features (stereo matching) along the epipolar lines. Rectification
consists in projecting the input images onto a common grid, given a reference altitude, such that the epipolar direction is
linewise. Consequently, any point having a different altitude than the reference altitude will experience a disparity in line di-
rection. Rectified and reference images are linked via the homography transforms $H_1$ and $H_2$. s2p calculates the homographies
by exploiting the RPC camera models. The stereo rectified images are fed to the stereo matching algorithm to find all pairs
of matching cloud features (i.e. image points (h,k) and (h+dx,k+dy), dx and dy being the disparities along x and y oriented
as the red arrows in Fig. 8 show, with dy=0 being the epipolar lines horizontal). The latter are linked by horizontal yellow
lines (see Fig. 8, top right). The block matching algorithm "More Global Matching" (MGM) (Facciolo et al., 2015) that we





use was selected among those available (Semi Global Block Matching - SGBM, Multi-Scale Multi-Window Stereo Matching - MSMW - Buades and Facciolo, 2015 - and so forth) as one of the top-ranked stereo vision algorithms. MGM looks for the disparity map that minimizes an energy cost function. The disparity dx associated to each tie point is the distance between

290 two corresponding points in the rectified images (see Fig. 8, step 3). Cloud pixels closer to the camera (i.e. cloud top) having larger difference in relative shift along x in the two rectified images are associated to larger disparity values (deep red points), whereas points farther away from the camera (associated to lower difference in relative shift) are associated to lower disparities (light blue/brown pixels). Each pair of tie points is mapped back in the original images by homography inversion (see Eq. 5 and 6).

$$(r,c) = H_1^{-1} \oplus (h,k) \tag{5}$$

$$\left(r^{'},c^{'}\right) = H_2^{-1} \oplus (h+dx, k+dy) \tag{6}$$

The height of a point from ground $H_G$, given its location inside two images ((r,c) and (r',c')), is calculated iteratively via

RPCs. (r,c,$H_G$) and (r',c',$H_G$) are mapped into geographic coordinates (Lat, Lon,$H_G$) via the inverse RPCs model, eq. 7 and 8 and then are turned into UTM X,Y,Z coordinates. As a result a point cloud is generated. An array of shape [h, w] (h and w being image height and width in pixels of the rectified reference image), where each pixel contains the UTM X, Y, and Z coordinates of the triangulated point is obtained (see triangulated Z in Fig. 8, step 4). Moreover, s2p associates to each retrieved point the color coordinates [R, G, B] of the corresponding pixel from Rec ref(gray scale). A radiance threshold (about 2% of

the maximum radiance) is used for filtering deep dark points where no cloud pixels are present. This array of spatial and color coordinates is what we refer to as the ST point cloud.

$$Lon = \frac{p_5\left(r,c,H_G\right)}{p_6\left(r,c,H_G\right)} \tag{7}$$

$$Lat = \frac{p_7\left(r,c,H_G\right)}{p_8\left(r,c,H_G\right)} \tag{8}$$



**Figure 8.** Step-by-step 3D reconstruction via s2p algorithm. Top left - (1) reference and secondary images (Ref , Sec), Bottom left - (2) rectified images (Rec ref, Rec sec), top right - (3) tie points from block matching, 3.1) disparity map, 4) 3D reconstructed point cloud (z - altitude from ground).

The stereo retrieval from triplets, that we discuss in this paragraph, works by pairs. Given a set of three images let us assume Image 2 is our reference image. Tie points are first searched between pair (1,2) and then between pair (2,3). The corresponding disparity maps (one for pair (1,2) and one for pair (2,3)) are transformed into height maps (distance from ground) which were calculated on the grid of the original reference image. For a given tie point we then have two values of $H_G$: $H_{G1}$ from pair 315 (1,2) and $H_{G2}$ from pair (2,3), respectively. The mean of $H_{G1}$ and $H_{G2}$ is taken if ($H_{G2}$-$H_{G1}$) is less than a fusion threshold (user chosen), otherwise the point is discarded. From this point the 3D retrieval from triplets goes on exactly as we have seen for two images: $r_n$, $c_n$ and $H_G$ are mapped to Lat, Lon, $H_G$ and eventually into UTM X, Y, Z.



One of the main questions in preparation for the mission concerns the choice of the baseline and whether retrieval from triplets, in comparison to retrieval from pairs, is significantly more accurate and if it leads to a larger number of retrieved points. The number of detected points was plotted as a function of the acquisition number, for two and three satellites for the cumulus case (see Fig. 10(a)). This was done for a baseline B of 150 km (Sats 1-2 and Sats 2-3) and 300 km (Sats 1-3) and for three satellites (Sats 1-2-3) with a fusion threshold of 30 m. The latter was chosen after having ascertained that alternative values did bring no significant difference in terms of retrieval. We clearly see that the number of detected points decreases with the distance from nadir (A5-A6) because of the decreasing resolution for tilted views. Concerning the distance between the satellites, the number of detected points is higher for satellites separated by 150 km (sat 1-2 or sat 2-3) than for satellites separated by 300 km (sat 1-3) as the matching features are easier to identify.

The threshold of 30 m leads to some points being discarded, which appears to slightly reduce the number of detected points, when using three satellites instead of two. It is important to emphasize that the s2p algorithm uses two-view stereo at a time and then merges these independent two-view stereo reconstructions into a single reconstruction. This is contrary to full multi-view stereo methods (e.g., which use the whole set of three-views simultaneously). Multi-view methods are widely used in computer vision due to the advantages they bring over the two-view stereo (Zhang et al. 2019). Using full multi-view stereo methods might lead to different results in terms of 3D reconstruction via three cameras, namely that the 3D cloud envelope retrieval can be more accurate and lead to more detected points, than when using only two views. However, this must still be confirmed.

## 4.2  Testing of the stereo retrieval against the ground truth point cloud via M3C2 algorithm

The comparison of two sets of point clouds is not trivial and several metrics can be used. We chose to use the Multiscale Model to Model Cloud Comparison (M3C2) algorithm (Lague et al., 2013) to test the stereo retrieval against the GT point cloud. M3C2 is available from the open source project 3D point cloud and mesh processing software CloudCompare (Cloud Compare). M3C2 allows computing signed and robust distances for each point of the reference cloud along the local normal as represented in the schematic of Fig. 9(a). For any given core point i of the reference cloud ($S_1$ in step 1), a normal vector $\vec{N}$ is defined by fitting a plane to the neighboring points within a radius D/2 from i. A cylinder of cross section d (d being the projection scale) and longitudinal length L whose main axis goes through i is then aligned along $\vec{N}$ in order to intersect $S_1$ and the compared cloud ($S_2$). The intercept of $S_1$ and $S_2$ with the cylinder defines two subsets of points. Projecting each of the subsets on the axis of the cylinder gives two distributions of distance. The mean of the distribution gives the average position of the cloud along the normal direction, $i_1$ and $i_2$. The local distance between the two clouds $L_{M3C2(i)}$ is then given by the distance between $i_1$ and $i_2$. The results discussed in the next section were obtained assigning a value of 100m to the normal scale D, d and L. This was done after having ascertained that no significant difference in the outcome was attained for different values. The output of M3C2 are the absolute value of the distance $L_{M3C2}$ and the normal components of $\vec{N}$. The displacement vector components $L_{M3C2x}$, $L_{M3C2y}$ and $L_{M3C2z}$ of the core points are then calculated as the simple product $\vec{L}_{M3C2}=L_{M3C2}\vec{N}$. The ST point cloud (output of s2p) was compared against the GT point cloud, defined in Sect. 3.3. Fig. 9(b) shows the ST(left) and GT(right) clouds for the deep convective case (top line) and the cumulus case (bottom line), respectively. The color associated to each point is the altitude from ground Z. Also shown is the M3C2 absolute distance (center) for the deep





convective case (top) and the cumulus case (bottom). Both retrievals correspond to acquisition A5 (close to nadir view). For both test cases the number of ST points is quite dense and most of detected points match with the GT point cloud.





**Figure 9.** (a)- Schematic of M3C2 working principle - from (Lague et al., 2013). (b) ST(left) and GT(right) point clouds for the deep convective case (top) and the cumulus case (bottom), M3C2 absolute distance in meters (center) for the deep convective case (top) and the cumulus case (bottom). The red arrow is oriented along the z axis.

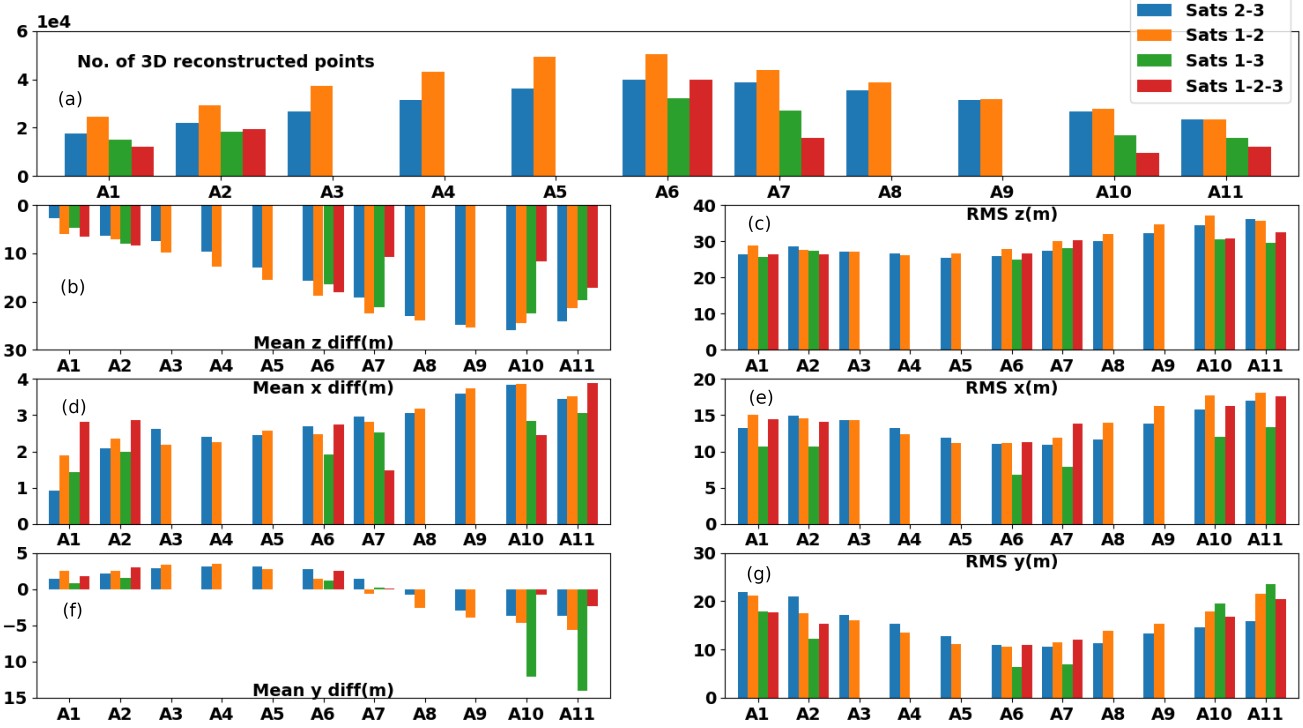

**Figure 10.** Results of the testing of the 3D retrieval for the cumulus case - Fig. 10(a): number of stereo retrieved points as a function of the acquisition number (A5, A6 are close to Nadir) for 2 vs 3 satellites and for B=150km (Sats 2-3 and Sats 1-2) and B=300km (Sats 1-3). Fig. 10(b), Fig. 10(d) and Fig. 10(f): mean difference between ST and GT point clouds for each acquisition along z, x, y. Fig. 10(c), Fig. 10(e) and Fig. 10(g): Root Mean Square Error (RMSE) of the difference between ST and GT point clouds for each acquisition along z, x, y.

To quantify the stereo retrieval error, for any given satellite configuration, we have measured the distance between retrieved
and ground truth points using M3C2. For each acquisition and from $L_{M3C2x}$, $L_{M3C2y}$ and $L_{M3C2z}$ we have then calculated the mean difference (bias) and the RMSE along the axes x, y and z. The three plots in Fig. 10(b), Fig. 10(d) and Fig. 10(f) show the mean difference along z, x and y axis as a function of the acquisition number. The mean difference (its absolute value) along z is less than 25 m while it is less than about 5 m and 15 m along x and y, respectively. The RMSE (see Fig. 10(c), Fig. 10(e) and Fig. 10(g)) for the z coordinate is less than 40 m and increases slightly with the distance from nadir. The same increase with
the view angle is more evident with $RMSE_x$ and $RMSE_y$. Similar results with bias < 4 m and RMSE < 25 m, were obtained for the deep convective cloud case (not shown here). Based on the s2p algorithm, it appears that none of the configurations significantly outperforms the others, in terms of retrieval quality. However, as already pointed out at the end of Sect. 4.1 these conclusions may differ, if analysis would use alternative 3D reconstruction algorithms, based on full multi-view stereo.



## 5 Inter-acquisition cloud development velocity from feature matching and 3D Stereo(ST) retrieval

In this section we look into the velocity retrieval method applied to the ST clouds (paragraph 5.1) and another independent method for deriving a ground estimate of cloud development velocity from the physical model (paragraph 5.2). As the retrieved ST point clouds are not sufficiently dense to be used for the matching of cloud features between two acquisitions, we look for tie points between successive images, by exploiting the block matching algorithm of s2p, and then map image points to 3D space points via 3D reconstruction.

### 5.1   3D cloud development velocity from 3D retrieved cloud envelopes

   The working principle of the ST velocity retrieval is illustrated in Fig. 11a. Tie points are first found from a pair of images taken from two consecutive acquisitions via MGM (see Fig. 11a, step 1, ST2A5 (top) and ST1A6 (bottom)). Red dots, representing the tie points so found, are linked by yellow lines (notice that only a few of them are shown given the high density of points). Each image from the chosen pair is then utilized with one or two simultaneous images (secondary images in the top right corner) to do the 3D retrieval of the cloud envelopes, as described in Sect. 4.1, for acquisitions A5 and A6, respectively. The tie

points from step 1 are then mapped in the rectified images (step 2) and the retrieved UTM X,Y and Z (step 3) are interpolated (to the nearest point if distance < 1 px) to the locations identified by the floating tie points (step 4). Ultimately, tie points in image ST2A5 ($i_5$, $j_5$) are mapped to tie points in space ($X_5$, $Y_5$, $Z_5$) for A5 and analogously, tie points in image ST1A6 ($i_6$, $j_6$) are mapped to tie points in space ($X_6$, $Y_6$, $Z_6$) for A6. The velocity vector is then derived according to eq. 9.

$$V_z = \frac{(Z_6 - Z_5)}{dt} \qquad V_x = \frac{(X_6 - X_5)}{dt} \qquad V_y = \frac{(Y_6 - Y_5)}{dt} \qquad (9)$$

   The error associated to the velocity retrieval is given by the contribution of the block matching and stereo retrieval errors. MGM mismatch error was quantified in a previous work (see Facciolo et al., 2015) in terms of bad pixel ratio (percentage of pixels with error > 1 px) and was compared to other state of the art semi-global matching methods, yielding the lowest average errors. In addition to false matches, rounding errors occur when the retrieved X, Y and Z (given on an integer grid) are

385 interpolated to the floating tie points coordinates. In paragraph 5.3, the retrieved velocity is compared to an estimate of cloud development velocity, derived from the GT envelopes, to which we turn next.



**ST cloud development velocity**

A5 → A6

1) Tie points (red dots) from A5 and A6 via MGM (Facciolo et al., 2015)
2) Tie points are mapped on to the rectified images
3) Triangulated 3D point cloud (z coord.) for A5 and A6
4) The triangulated point is interpolated over the tie points

$$STV_x = (x_6 - x_5)/dt$$
$$STV_y = (y_6 - y_5)/dt$$
$$STV_z = (z_6 - z_5)/dt$$

(a)

**GE cloud development velocity**

A5 → A6

1 Find Δx, Δy that minimize RMSE($z_5$, $z_6$) to account for advection Δx/dt, Δy/dt

2 Shift GTA6 an amount **Δx** and **Δy**

3 Measure M3C2 dist(GT5,GT6)

**GE velocity vector**

$$GEV_x = M3C2(x_6, x_5)/dt + \Delta x/dt$$
$$GEV_y = M3C2(y_6, y_5)/dt + \Delta y/dt$$
$$GEV_z = M3C2(z_6, z_5)/dt$$

(b)

**Figure 11.** Cloud development velocity algorithms: (a) from ST clouds - (1) tie points from images ST2A5 and ST1A6, (2) tie points are mapped on to the rectified images, (3) 3D retrieved point clouds (for A5 and A6), (4) 3D point clouds interpolated to the tie points, (5) mean ST velocity vector. (b) From GT clouds - (1) Δx, Δy that minimize RMSE($z_5$,$z_6$) are calculated, (2) GTA6 is translated an amount Δx, Δy to account for advection, (3) M3C2 cloud to cloud distance is calculated, (4) The mean GT velocity vector is derived.





## 5.2 3D cloud development velocity from the GT point clouds: Ground Estimation(GE)

3D cloud development velocity was derived from the GT point clouds from two consecutive acquisitions. This is further illustrated in Fig. 11b for acquisitions 5 and 6. First we determine the horizontal displacement that minimizes the RMSE($Z_{A5}$,

$Z_{A6}$) with Z being the altitude from ground (step 1). This is done to account for the advection which we assume to be constant, over the extent of the cloud field, for this test case. We find that the superposition of $GT_{A5}$ (reference cloud) and $GT_{A6}$ (secondary cloud) is optimized by shifting $GT_{A6}$ an amount $\Delta x$=130 and $\Delta y$=120 m (step 2). Then the M3C2 distance, discussed in Sect. 4.2, is used to measure cloud to cloud distance (step 3). As a reminder, this is the distance between two average positions, along the local normal, from the reference cloud to the secondary cloud. This distance is decomposed along

X, Y and Z to get the velocity vector after adding the average advection (see eq. 10) (step 4). The resulting Z component of the velocity GTVz is shown in the right corner of Fig. 11b. It should be specified that as opposed to the retrieved ST products, this method does not rely on finding matching cloud features. For any given point from the reference cloud, its matching point from the secondary cloud is searched for along the local normal.

$$V_z = \frac{M3C2(Z_{11}, Z_9)}{dt}, \quad V_x = \frac{M3C2(X_{11}, X_9) + \Delta x}{dt}, \quad V_y = \frac{M3C2(Y_{11}, Y_9) + \Delta y}{dt} \tag{10}$$

In the coming section we will compare the retrieved cloud velocity to the velocity derived from GE.





### 5.3 Comparison between the ST velocity and GE velocity

With respect to the following comparison, the velocity of each tie point, in 3D space, is compared to the velocity of the nearest point extracted from the GT velocity cloud, if the distance is less than 100 m. Fig. 12(a) and Fig. 12(b) show the cloud velocity vectors obtained between A5 and A6 from the ST and GT clouds, respectively. In the top part of each figure from left to right

each cloud pixel is associated to a color that represents the value of the velocity along z, x and y, respectively. The bottom part of each figure shows instead the distribution of the cloud development velocity. With regard to z, red (blue) color is associated to upward (downward) development. The cloud is mostly growing upwards with a mean velocity of 1.6 m/s and peaks up to 15 m/s in the uppermost part of the cloud (around line 2400 and column 1500). The horizontal velocity components show that the cloud is moving along the diagonal direction with a mean velocity of (6.5, 6.0) m/s. While in the x direction the

cloud moves more uniformly ($\sigma$ = 1.0 m/s), in the y direction the distribution is wider ($\sigma$ = 6.5 m/s) suggesting a diverging cloud development for the highest part of the cloud. With regard to the GE velocity derived from the GT envelopes, the mean velocities of 0.6 m/s, 6.5 m/s and 6.1 m/s are consistent with the ST mean velocities. This agreement, although more evident with respect to the horizontal velocities, confirms the expected vertical growth, for most pixels, and also the asymmetry seen in the cloud development along y. The distributions of the GEs are clearly narrower. However, we should remember that the

M3C2 distance is not a measure of the distance between matching cloud features but the distance between two average points along the local normal from cloud to cloud. As a result, the M3C2 distance is an underestimation of the actual distance travelled by matching features as, for instance, at the top of an eddy where cloud development does not occur along the local normal. The double modes in the retrieved Vy histogram, not present in the GE, could be associated to the divergence of the very cloud top in the center right part of the image. However, further analysis is required to confirm that this is not due to an artifact.

**Figure 12.** (a) STV from A5-A6. Top part from left to right: the components z, x and y of the retrieved velocity STV. Bottom Part from left to right: z, x, y distributions. (b) Same as in (a) but for GTV.





Similar results (not shown) have been obtained for other pairs of acquisitions with small differences for the mean values
of the velocity and larger dispersion for the ST velocities in comparison to the ground estimate. As discussed previously, the
two methods have their own sources of error with the retrieval relying on image data whereas the GT velocities are ultimately
derived from the physical model. Furthermore, while the retrieval method is based on finding matching points, the GT method
relies on measuring cloud to cloud distance along the local normal. These two elements certainly contribute to the discrepancy
observed. This can be reduced by tracking the position of a group of tie points over several acquisitions, to which we turn next.

### 5.4    Velocity vectors from ST and GT point clouds via point tracking

Following the idea mentioned at the end of the previous paragraph, an alternative way for comparing the ST and GT velocities
consists in taking the slope of the regression line (over several acquisitions) of the mean X, Y, Z for a given set of tie points.
In what follows, seven clusters of tie points, from different cloud cells, each identified by a specific colour as in Fig. 13a, were
tracked from A1 (far from nadir) to A6 (close to nadir) by applying MGM every two acquisitions (A1=>A2, A2=>A3, ...,
A5=>A6). For each cluster of points and for each acquisition, 3D space coordinates are retrieved via image-to-space mapping
through 3D reconstruction, as explained in Sect. 4.1. The same sets of points are shown in 3D space (top view, X-Y plane) in
Fig. 13b together with the GT point clouds (grey points) for each acquisition. Each group of points, with space coordinates (X,
Y, Z), is contained in the rectangular region defined by $(X_{min}, X_{max})$ and $(Y_{min}, Y_{max})$ that is shown in the figure inset (light
blue rectangle) for the red points of acquisition A6. The same light blue area was divided into 10x10 atmospheric columns
(notice red grid) from each of which the uppermost point of the GT point cloud was extracted (black points - GT max). This
is justified by the visibility of the GT cloud top. We compare the mean value of Z for the red points (ST points) to the mean
value of Z for the black points (GT points). In Fig.14, the mean ST Z and mean GT Z values are plotted as a function of the
acquisition (time) together with the regression lines and SD as error bars. The slope of the regression lines for each cluster
gives the mean vertical velocity over the six acquisitions. The agreement between mean ST Z and mean GT Z is good (within
the error range) and shows that the cloud development velocity can be estimated with good accuracy over a time interval of 100
s for instance, as it was done in this case. Lastly, it should be noticed that with this second approach we can compensate for
the attitude errors of the platforms. However, these sources of error have not been taken into account to carry out this work that
instead is based on realistic but perfect images, "perfect" in that neither radiometric nor attitude errors have been accounted for
as out of the scope of this paper.



**Figure 13.** (a) Image tie points tracked from simulations A1 to A6 via MGM. Each group of points is associated to a given colour. (b) ST tie points from (a) in 3D space and GT point clouds from A1 to A6. Zoomed-in inset showing the atmospheric columns for point selection from the GT point cloud.





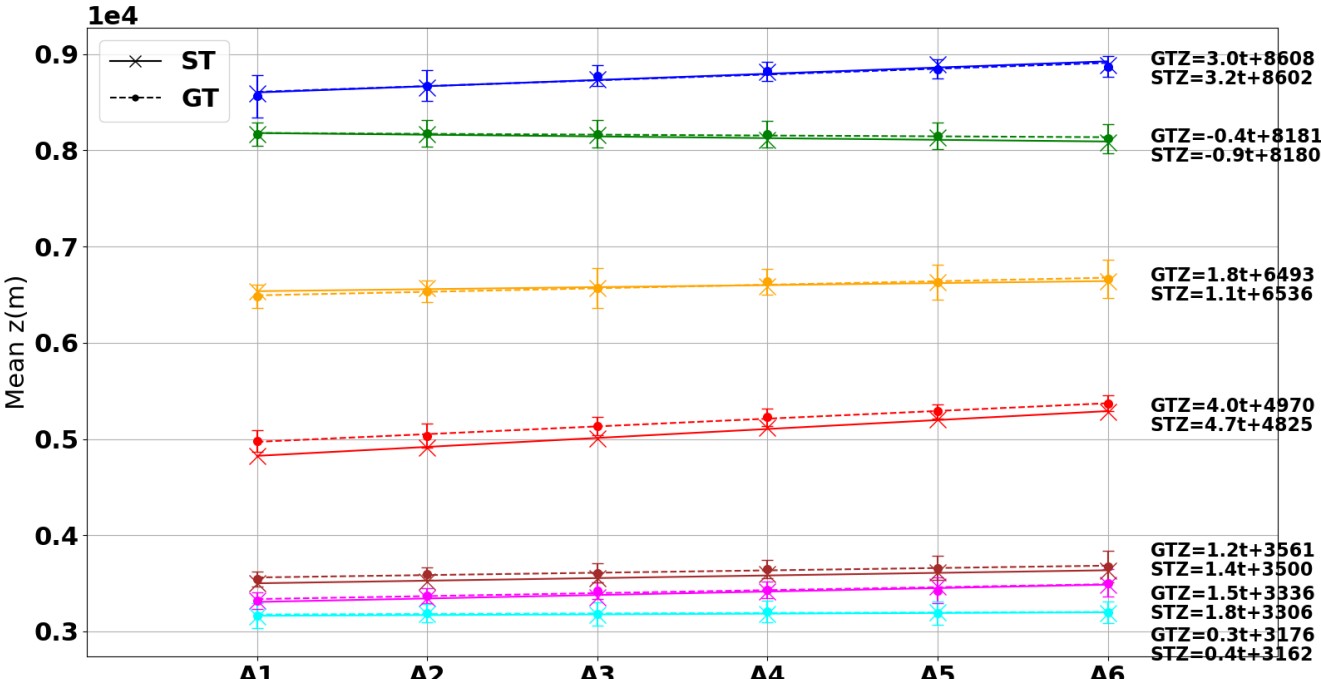

**Figure 14.** Time series of mean ST Z and mean GT Z.

## 6 Conclusions

A method to retrieve the 3D cloud envelope and 3D cloud development velocity from satellite imagery was presented. As actual data from the CLOUD cameras of the C3IEL mission are not yet available, multi angular simulated images were used. These were obtained via Meso-NH and the radiative transfer code 3DMCPOL for a deep convective cloud case and via SAM and the
image renderer Mitsuba for a shallow cumulus case. Such simulations are realistic in that they were obtained accounting for the geometric camera model while images are considered "perfect" in that neither radiometric nor geometric noises were taken into account. Cloud geometry retrieval, from pairs or triplets of images, was attained via 3D stereo reconstruction and was tested against the reference point cloud derived from the physical properties output of the cloud models. The error associated to the cloud geometry retrieval, quantified in terms of RMSE (<40 m) and Bias (<30 m), was obtained by measuring cloud-to-cloud
distance between retrieved and reference point cloud. The inter-acquisition cloud velocity was instead retrieved by identifying matching cloud features from two consecutive acquisitions (for instance A1, A2) of the same cloud scene via MGM. The 3D reconstructed point cloud, for each acquisition, is then used to map the matching cloud features from image to space. The inter-acquisition velocity vector, derived from the retrieved space coordinates $(X_1, Y_1, Z_1, X_2, Y_2, Z_2)$, is the ratio of the space traveled by each tie point to the inter-acquisition time step (20 s). The results were compared against a ground estimate of
cloud development velocity that was obtained via a completely independent method that relies on minimising the RMSE of the reference cloud top between two acquisitions and on measuring cloud to cloud distance. The mean velocities retrieved from the images and from the Ground Estimation are in close agreement although the ST velocities, relying on matching cloud features from two consecutive acquisitions, are more dispersed than the GE velocities derived instead from cloud-to-cloud distance. A more sophisticated assessment for the stereo velocity would require applying the same tracking method, used for the images, to
track the movement of identified cloud structures from the GT point clouds. In the meantime we developed an easy workaround that consists in tracking a cluster of points over several acquisitions. The slope of the regression line to the mean Z position of a set of matching cloud features, identified over several acquisitions, gives the mean velocity. This method was applied to the point clouds derived from the images and to those derived from the physical model. Results from this second comparison show that an estimate of the stereo velocity can be achieved with good accuracy (within the error range).


*Code and data availability.* Data are available upon request. The stereo retrieval algorithm s2p used to carry out this work is an open freeware software.




*Author contributions.* P.D. and C.C. conceived the original idea. Da.R., R.B., Y.S., Di.R. helped supervise the project. C.C. and V.H. carried out the image simulations. Di.R. and V.H. provided the output of the physical cloud model. R.B., L.F. and V.H. did the modeling of the
geometric camera models. P.D. performed the computations and tested the retrieval methods. C.C. and Da.R. supervised the findings of this work. All authors discussed the results and contributed to the final manuscript. P.D. wrote the manuscript with support from C.C, Di.R., R.B. and Y.S.. All authors provided critical feedback and helped shape the research, analysis and manuscript.

*Competing interests.* No competing interests are present.

*Acknowledgements.* This work was supported by CNES through the TOSCA program under grant no. 5414/MTO/4500065502 and no. 6728/MTO/4500069332. The CNES is also acknowledged for the first author contract support. This research was funded by the Israel Ministry of Science and Technology (MOST), grant number 2-17377.





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
