# Peer review of "3D cloud envelope and cloud development velocity from simulated CLOUD/C3IEL stereo images"

_Atmospheric Measurement Techniques, 2022_

## Author Comment (AC2)

Dear AMT Editors and Reviewers,

Thanks for starting the review process and for the precious comments and suggestions provided. In what follows we list the comments expressed by the Anonymous Referee #1 (paragraphs written in blue color), provide our answers (paragraphs written in black color) to the raised points and explain how we addressed each point in the revised manuscript.

Comment on amt-2022-61 from Anonymous Referee #1
Referee comment on "3D cloud envelope and cloud development velocity from simulated CLOUD/C3IEL stereo images" by Paolo Dandini et al., Atmos. Meas. Tech. Discuss., https://doi.org/10.5194/amt-2022-61-RC1, 2022

This work proposes a method to estimate cloud envelope and cloud development velocities based on high resolution satellite train imagery. Because the planned satellite train is not launched yet, the work relies on simulated image data to show the proof of concept. Although the simulated data are perfect in the sense that noise is not accounted for, they are helpful to compare against the ground truth (that will not be available in the case of real data) and therefore show the potential. Before publication, I suggest that the following points are addressed:

Line 12-14: "An independent method based on optimizing the superposition of the cloud top, issued from the atmospheric research model, allows to obtain a ground estimate for the velocity from two consecutive acquisitions." I do not understand this statement, what is meant by superposition in this context? Maybe it is more clear in the text but it just makes the abstract confusing.

Authors' reply :
For more clarity, in the abstract, lines 12-14 (pp. 1) (lines 15-16 – pp. 1 in the revised manuscript), we replace the following sentence :
«An independent method based on optimizing the superposition of the cloud top, issued from the atmospheric research model, allows to obtain a ground estimate for the velocity from two consecutive acquisitions. »
with the following one :
 «An independent method based on minimizing the RMSE for a continuous horizontal shift of the cloud top, issued from the atmospheric research model, allows to obtain a ground estimate of the velocity from two consecutive acquisitions.»

Comment on amt-2022-61 from Anonymous Referee #1
Similar for the lines 14-17, please considering restating these lines to make it easier for the reader to understand.
Authors' reply:
For more clarity, in the abstract, lines 14-17 (pp.1) (lines 18-21 – pp. 1 in the revised manuscript), we replace the following sentence:
«The distribution of retrieved velocity and ground estimate exhibits small biases but significant discrepancy in terms of distribution width. Furthermore, the average velocities derived from the mean altitude from ground for a cluster of localized cloud features identified over several acquisitions, both in the simulated images and in the physical point cloud, are in good agreement.»

with the following one:
«The mean values of the distributions of the stereo and ground velocities exhibit small biases. The width of the distributions is significantly different with higher distribution width for the stereo retrieved velocity. An alternative way to derive an average velocity over 200s, which relies on

tracking clusters of points via image feature matching over several acquisitions, was also implemented and tested. For each cluster of points, mean stereo and ground positions were derived every 20s over 200s. The mean stereo and ground velocities, obtained as the slope of the line of best fit to the mean positions, are in good agreement.»

Line 56, Do these error bands differ by magnitude of the velocity and the height? If so, is it better to give a percentage?
Authors' reply:
The error assessment was done in the work by Horvath and Davis (Horvath and Davis, 2001a) who demonstrate the feasibility of the retrieval of cloud top and winds from MISR. The errors were determined on simulated data and reported as absolute value. We prefer to report the errors as the authors of the work did. However, we correct the sentence by referencing the work, from the same authors, that presents the first retrievals from actual data (Horvath and Davis, 2001b).

We then correct the following sentence, lines 55-56 (pp. 3) (lines 64-66 – pp. 3 in the revised manuscript):
«The first retrievals from actual data (Horvath and Davis, 2001) were consistent with the prelaunch error estimates of ±3 m/s and ±400 m for winds and heights, respectively.»
by replacing it with the following one:
«The first retrievals from actual data (Horvath and Davis, 2001b) were consistent with the prelaunch error estimates (Horvath and Davis, 2001a) of ±3 m/s and ±400 m for horizontal winds and heights, respectively.»
We also add the reference to the work of Horvath and Davis (Horvath and Davis, 2001b) in the edited manuscript.

We also add after the following sentence, lines 56 (pp. 3) (lines 66 – pp. 3 in the revised manuscript):
"These retrievals were obtained for the first time from the polar orbiting spacecraft Terra."
what follows:
"Only recently, Mitra et al. (2021) provided the first evaluation of the Terra Level 2 cloud top height (CTH) retrievals against the Cloud-Aerosol Transport System (CATS) Lidar CTHs, with uncertainties of −280 ± 370 m."

and then correct the following phrase:
"The main limitations of their method is the fact that vertical cloud motion is neglected and a constant horizontal cloud advection over the domain is assumed which under intense convection, for instance, may lead to unreliable retrieved winds."
as follows:
"The main limitations of the method of Horvath and Davis is the fact that vertical cloud motion is neglected and a constant horizontal cloud advection over the domain is assumed which under intense convection, for instance, may lead to unreliable retrieved winds."

Consequently, we also add the work by Mitra et al. (2021) to our list of references.

Line 75, Please be specific, magnitude acceleration in what? Computation time?
Authors' reply:
We rephrase the following phrase, lines 74-75 (pp. 3) (lines 85-86 – pp. 3 in the revised manuscript):
«Sde-Chen et al. (2021) devised a neural network for spaceborne 3D cloud CT, leading to 5-order of magnitude acceleration, relative to Levis et al., 2015.»

by rewriting:

«Sde-Chen et al. (2021) devised a neural network for spaceborne 3D cloud CT, leading to a significant reduction in terms of run-time, relatively to Levis et al., 2015.»

Comment on amt-2022-61 from Anonymous Referee #1
Line 101, Do I understand correctly that the image will not be taking images continuously but will start when triggered and for 200 seconds only. If so, please make this explicit and explain what will trigger the image capture event.
Authors' reply:

Line 101, that is correct. The cameras will not be taking images continuously. When triggered, the cameras will take two or three simultaneous images every 20 seconds during 200 seconds (11 acquisitions=11 pairs or triplets of simultaneous images). While no dynamic triggering is foreseen, the acquisition sequence will be scheduled at selected latitudes, depending on the season and on climatology, where and when clouds are more likely to be observed. Moreover, the acquisition schedule will be periodically (two/three times a year) updated to target the observation of convective cloud scenes and, when possible, to achieve co-location with ground observations. However, with such acquisition strategy, measurements over clear skies will not be avoidable. As for the synchronization of the image capture event from the different cameras, the pulse per second (PPS) signal from the GNSS receiver will allow to achieve atomic-clock accuracy with no need of communication between satellites.

For more clarity we correct the phrase, lines 102-105 (pp. 4) (lines 113-116 – pp. 4 in the revised manuscript):

"The observational strategy for the imagers will consist in multi-angular observations of a given cloud scene during 200 s with instantaneous stereoscopic pairs or triplets captured every 20 s (11 acquisitions A1-A11 see Fig. 1) corresponding to the life time of cloud perturbations at small scale."
as follows:

"The observational strategy for the imagers will consist in multi-angular observations of a given cloud scene during 200 s with instantaneous (not continuous) stereoscopic pairs or triplets captured every 20 s (11 acquisitions A1-A11 - see Fig. 1) corresponding to the life time of cloud perturbations at small scale. 3 or 4 sequences of acquisition, each of the duration of 200s, will be acquired per orbit. The image capture event will not be triggered dynamically but scheduled at specific latitudes, depending on the season and on climatology, where and when clouds are more likely to be observed. This schedule will also be tuned periodically, two/three times a year, to maximize the chance of observing convective cloud scenes and to achieve co-localized measurements with ground observations, when possible. As for the synchronization of the image capture event from the different cameras, the pulse per second (PPS) signal from the GNSS receiver will allow to achieve atomic-clock accuracy with no need of communication between satellites."

For more clarity we also add in the caption of Figure 1:
"Instantaneous (not continuous) stereoscopic pairs or triplets captured every 20 s over 200 s, that is 11 acquisitions A1-A11."

Comment on amt-2022-61 from Anonymous Referee #1
Line 110, How accurately can the satellite positions can be retained? That is what are the bounds of change in baseline? Is this likely to impact the retrieval quality?

Authors' reply:
Satellite positions are assumed to be known with good accuracy thanks to the GNSS receiver on-board. Satellite relative position mainly depends on the frequency of the operations of adjustment, coupled with solar activity, that for C3IEL are targeted to guarantee a bound of change of about 50

km, at worst. Rather than simulating the effect of "small" baseline changes to get insights into the feature matching quality and reconstruction uncertainty, we used two baselines, 300 km and 600 km, and found no significant difference in terms of stereo-retrieval, as Figure 10 shows. Our view is that the pointing error for an orbit of 600 km affects the line-of-sight (LOS) much more severely than a small error (as small as around 10 meters) on the satellite position. However, as for the work here presented neither orientation nor position errors were accounted for. This should be done in the future with the aim of decoupling cloud motion from the contribution of AOCS (Attitude and Orbit Control System) errors.

Lines 443-445 (pp. 30) (lines 501-503 – pp. 32 in the revised manuscript), we rewrite the following sentence:
"However, these sources of error have not been taken into account to carry out this work that instead is based on realistic but perfect images, "perfect" in that neither radiometric nor attitude errors have been accounted for as out of the scope of this paper."
as follows:
"However, these sources of error and likewise the satellite position error, have not been taken into account to carry out this work that instead is based on realistic but perfect images, "perfect" in that neither radiometric nor attitude errors have been accounted for as out of the scope of this paper."

We also add in the conclusions (last paragraph) that:
"However, in the future, to generalize our results, we plan to test our methods for other cloud types, scenes and solar geometries. This will be done by taking into account radiometric noise and image distortion as well as satellite orientation and position errors. This will allow to quantify the degradation of the results here obtained for "perfect simulations"."

We also add in the introduction of section 3 (lines 165-167 – pp. 9 in the revised manuscript).
"This second simulation is thus more realistic than the first one and in the future will allow to account for image distortion and satellite orientation error."

Line 200 (pp. 12) (line 273 – pp. 16 in the revised manuscript), after:
"In this way, the sphericity of the orbit and the orientation of the satellites are accounted for."
we add:
"In this work, satellite position and orientation are assumed to be known exactly. However, this is not true and the corresponding errors are expected to deteriorate the results here presented. We will be able to test such statement, in the future, once these sources of error will have been modeled for each camera, by exploiting the combined use of Euclidium and 3DMCPOL."

Comment on amt-2022-61 from Anonymous Referee #1
Line 140, is 22.5 km the location from a reference point? Please clarify.
Authors' reply:
For more clarity, we rewrite the following sentence, lines 140-141 (pp. 9) (lines 221, 222 – pp. 13 in the revised manuscript):
«Fig. 4 shows the vertical section, at 22.5 km, of total water content, as well as vertical and horizontal wind components.»
as follows:
(Notice that Figure 4 becomes Figure 6 in the revised manuscript!)
«Fig. 6 shows the vertical section, at 22 km (6 km from the origin of the y axis located at 16 km, see Fig. 6g,6h), of total water content, as well as vertical and horizontal wind components.»
We also edit the caption of Figure 6:
"Deep convective cumulus physical properties. Fig. 6a, 6b, 6c: vertical section (y=22.5 km) of cloud total water content, vertical and horizontal wind components, respectively - Fig. 6d, 6e, 6f: vertical section (y = 22.5 km) of total extinction coefficient, cloud phase and effective radius,

respectively. Value of 31 (dark red) is associated to voxels where the mean ice phase function is used - Fig. 6g, 6h: liquid optical thickness and total optical thickness (liquid + ice), respectively."
as follows:
"Deep convective cumulus physical properties. Fig. 6a, 6b, 6c: vertical section (y=22 km, that is 6 km from the origin of the y axis located at 16 km) of cloud total water content, vertical and horizontal wind components, respectively - Fig. 6d, 6e, 6f: vertical section (y = 22 km) of total extinction coefficient, cloud phase and effective radius, respectively. Value of 31 (dark red) is associated to voxels where the mean ice phase function is used - Fig. 6g, 6h: liquid optical thickness and total optical thickness (liquid + ice), respectively."

Comment on amt-2022-61 from Anonymous Referee #1
Why do Shallow cumulus clouds have better resolution? Shouldn't it be the opposite?
Authors' reply:
The spatial resolution depends on the configuration of the atmospheric research models. As these simulations are highly time demanding, we used a deep convective cloud case, modeled via Meso-NH, from previous work (Strauss et al, 2019).
The atmospheric research model SAM used to simulate the trade wind cumulus was instead configured with a resolution of 20m.

Comment on amt-2022-61 from Anonymous Referee #1
Lines 327-333, is there a specific reason for merging 2xtwo-view instead of using three-view when data from Sats 1-2-3 are used?
Authors' reply:
As we stated in the manuscript, the working principle of the s2p algorithm is based on stereo matching. Stereo Matching and Structure from Motion (SfM) are currently the predominant methods to derive geometric structures from satellite images (de Franchis et al., 2014; Zhang et al.,2019). However, while SfM reconstructs information of multiple images, Stereo Matching is restricted to single image pairs. SfM based approaches are inherently better suited to process large (unstructured) image sets such as multi-date satellite imagery and could be tested in future works. We clearly stated (lines 328-333, pp. 21) (lines 374-379 – pp. 22 in the revised manuscript) that:
«It is important to emphasize that the s2p algorithm uses two-view stereo at a time and then merges these independent two-view stereo reconstructions into a single reconstruction. This is contrary to full multi-view stereo methods (e.g., which use the whole set of three-views simultaneously). Multi-view methods are widely used in computer vision due to the advantages they bring over the two-view stereo (Zhang et al. 2019). Using full multi-view stereo methods might lead to different results in terms of 3D reconstruction via three cameras, namely that the 3D cloud envelope retrieval can be more accurate and lead to more detected points, than when using only two views.».

We rephrase the last sentence of sec. 4.1, lines 331-333 (pp. 21) (lines 379-382 – pp. 22 in the revised manuscript) as follows:
"Using full multi-view stereo methods might lead to different results in terms of 3D reconstruction via three cameras, namely that the 3D cloud envelope retrieval can be more accurate and lead to more detected points compared to when using two views."

Comment on amt-2022-61 from Anonymous Referee #1
Figure 10, Do I understand correctly from the figure that no points are retrieved with Sats 1-3 and Sats 1-2-3 scenarios in A3-5 and A8-9 views? If so, how can you say that none of the configurations outperform? Also, what is thee reason for the skewed-towards-A9/A10-views distribution of error in z in Figure 10.b?
Authors' reply:

Figure 10: For more clarity we repeated the calculations and accordingly updated figure 10 that now includes calculations for all scenarios. The conclusions are the same as previously, that is none of the configurations outperforms the others.

However, by repeating the calculations, we notice that the mean Y error for configurations 1-3 becomes now 5 m/s whereas previously it was about 15 m/s. We then correct the sentence, lines 357, 358 (pp. 24) (lines 406, 407 – pp. 25 in the revised manuscript):

«The mean difference (its absolute value) along z is less than 25 m while it is less than about 5 m and 15 m along x and y, respectively.»

as follows:

«The mean difference (its absolute value) along z is less than 25 m while it is less than about 5 m along x and y.»

and then we add:

"Such values can partly be ascribed to the stereo-opacity bias, associated to low extinction near the cloud top,  as discussed by Mitra et al. (2021)."

In the abstract we also correct, lines 9-11 (pp. 1) (lines 12-14 – pp. 1 in the revised manuscript):

"The accuracy of the retrieval of cloud topography is quantified in terms of RMSE and bias that are respectively, less than 25 m and 15 m for the horizontal components and less than 40 m and 25 m for the vertical component."

as follows:

"The accuracy of the retrieval of cloud topography is quantified in terms of RMSE and bias that are respectively, less than 25 m and 5 m for the horizontal components and less than 40 m and 25 m for the vertical component."

Concerning the skewed distribution of the error along z for the A9-A11 views, we add the following sentence, line 359, (pp. 24) (lines 406, 407 – pp. 25 in the revised manuscript) after 'The mean difference (its absolute value) along z is less than 25 m while it is less than about 5 m along x and y.':

"The skewed distribution of the error in Figure 10b, for the views A9-A11, may be due to the fact that fewer cloud features are visible as the clouds are less illuminated by the sun, with larger portions of the cloud field shaded, as it can be seen from Figure 5g, 5h and 5i."

Still with respect to the calculations concerning the cumulus cloud, but this time with regard to Figure 9b, for better clarity and to improve on points visibility, we replace the three figures at the bottom of Figure 9b showing the stereo and ground truth cumulus clouds and the corresponding M3C2 cloud-to-cloud distance.

Comment on amt-2022-61 from Anonymous Referee #1
Equation 10, why are the subscripts suddenly 9 and 11?
Authors' reply:
Equation 10, we replaced 9 and 11 with 5 and 6, respectively.

Comment on amt-2022-61 from Anonymous Referee #1
Line 418, please explain why a dual mode caused by a possible "divergence of the cloud top in the center right part" would not show in the GE distribution?
Authors' reply:
This diverging effect appears in both the retrieved velocity and the GE velocity although it is more pronounced in the former. The absence of a dual mode in the GE distribution is due to the fact that the M3C2 method underestimates the actual distance vector especially when cloud development does not occur along the local normal as, for instance, at the top of an eddy.
Lines (418, 419 – pp. 28) (lines 473-476 – pp. 30 in the revised manuscript), we rephrase the sentence :

"The double modes in the retrieved Vy histogram, not present in the GE, could be associated to the divergence of the very cloud top in the center right part of the image."
as follows:
"The double modes in the retrieved Vy histogram, which could be associated to the divergence of the very cloud top in the center right part of the image, are not present in the GE distribution. Although a hint of cloud divergence is also visible in the ground estimate of Vy, the double modes are smoothed out because of the underestimation of the distance vector associated to the use of the M3C2 metric."

---

## Author Comment (AC3)

Dear AMT Editors and Reviewers,

Thanks for starting the review process and for the precious comments and suggestions provided. In what follows we list the comments expressed by the Anonymous Referee #2 (paragraphs written in blue color), provide our answers (paragraphs written in black color) to the raised points and explain how we addressed each point in the revised manuscript.

Comment on amt-2022-61 from Anonymous Referee #2
This paper describes a method to estimate the 3D cloud envelope and development velocity using simulated images of a triplet of small satellites in a sun-synchronous orbit at 600 km height. The focus lies on trade wind cumulus and deep convection, while the methodology relies on stereo analysis and tracking to compute the 3D points of the cloud envelope and subsequent cloud motion estimation. The study assesses the feasibility and accuracy of the proposed mission design.

The paper is well written and structured and the study is thoroughly conducted and the topic is scientifically relevant. I suggest publication after the following points have been addressed:

1. Fig. 1:
Could be a bit more detailed. Just a few more numbers and lines if possible. For example the height of and the distance between the individual satellites might be good for directly understanding the geometric setup, which is important for stereo analysis and triangulation.

Authors' reply:
We edit the figure by adding a line with arrows that specifies the height of the satellites from ground and additional lines that indicate the baseline in case of two and three satellites.

Comment on amt-2022-61 from Anonymous Referee #2
2. Line 106:
"consequently simultaneously". Sounds a bit strange. Maybe just leave the "consequently" out.

Authors' reply:
Line 106: We remove "consequently" so that the phrase (lines 105-107, pp. 4) (lines 122-124 – pp. 5 in the revised manuscript) now becomes:
"The measurements of these space-borne sensors will simultaneously document the vertical cloud development retrieved by a stereoscopic method, the lightning activity and the distribution of water vapor at a high resolution by exploiting the multi-angle acquisitions."

Comment on amt-2022-61 from Anonymous Referee #2
3. Was it described somewhere what axis the along-track direction was (x or y)? Maybe add it to one of the initial figures so that analysis later is easier.

Authors' reply:
We now add x (along track) and y (across track) arrows in both Figure 2 and Figure 6 (that becomes Figure 5 in the revised manuscript).
In this respect we correct the phrase, lines 114,115 (pp. 7) (lines 132, 133 – pp. 7 in the revised manuscript):

"It should be noticed that satellite is moving from North to South and while the across track resolution remains almost constant the along track resolution is increasing for tilted views."

and rewrite as follows:

"It should be noticed that the satellites are moving from North to South (see x arrow in Figure 2) and while the across track resolution remains almost constant the along track resolution is decreasing for tilted views."

We also add in the caption of Figure 2 and Figure 6 (that becomes Figure 5 in the revised manuscript) the following sentence:

"The along and across track directions are identified by the x and y arrows, respectively."

4. Line 115 + 116:
For a tilted view shouldn't the track resolution be decreasing as a larger area is projected to a smaller image area? The same with the Ground Sampling Distance. Shouldn't it increase for a tilted view? Maybe just a misunderstanding.
Authors' reply:
That is correct. Thanks for pointing this out.
We correct the following sentence (lines 114-116, pp. 7) (lines 132-134 – pp. 7 in the revised manuscript):

"It should be noticed that satellite is moving from North to South and while the across track resolution remains almost constant the along track resolution is increasing for tilted views. This leads to an increase of the footprint in the along track direction and a reduction of the ground sampling distance (GSD)."

and rewrite as follows:

"It should be noticed that the satellites are moving from North to South (see x arrow in Figure 2) and while the across track resolution remains almost constant the along track resolution is decreasing for tilted views. This is due to the increase of the footprint in the along track direction and the consequent increase of the ground sampling distance (GSD)."

5. Line 185:
I think the field of view is always constant as it depends on the camera. If the angle representing the image projection of the ground area is meant then it should be smaller with tilted view, shouldn't it?

Authors' reply:
Lines 184,185 (pp. 12) (lines 266, 267 – pp. 16 in the revised manuscript), we correct the following phrase:
"Furthermore, pixel size increases with the distance from the image center and so does the field of view for tilted views."

by re-writing as follows:

"Furthermore, pixel size increases with the distance from the image center and so does the footprint for tilted views."

6. Line 278:
You write that the cameras are affine for a small tile of an image. Considering the field of view in this simulation of 1 degree, the cameras already are very weakly perspective. Does an additional tiling matter?

Authors' reply:
The s2p pipeline was conceived for 3D stereo processing of pushbroom camera images that are not rectifiable as opposed to images taken from pin-hole cameras. Errors in the rectification may result in a vertical disparity (epipolar error) between corresponding points in the rectified images, which may significantly affect the performance of the stereo matching. By approximating the sensor by an affine camera model on small image tiles leads in practice to an almost perfect rectification, with a very small epipolar error.
However, in our case, although simulations are weakly perspective, especially true when the relative distance (scene depth) between two points of a 3D object along the optical axis is much smaller than the average distance to the camera, we do not need further tiling as rectification works well (with no epipolar error) by setting the tile size equal to the ROI (865px x 865px).

For more clarity, after the following sentence (lines 277-278, pp. 18) (lines 314, 315 – pp. 18 in the revised manuscript):

"The s2p process pipeline can be summarized as follows: input images are first cut into tiles, where the cameras are assumed to be affine (i.e. perspective projection)."

we add:

"With respect to the calculations presented in this work, we use a tile size equal to the ROI (865px x 865px) as we achieve satisfactory rectification with no need of further reduction of the tiling."

7. Line 279:
Maybe shortly describe what an epipolar line is and why it's useful.
Authors' reply:
Lines 278, 279 (pp 18) (lines 316-318 – pp. 18 in the revised manuscript), we edit the following sentence:
"With regard to Fig. 8, the input reference (ref) and secondary images (sec) are first rectified (rec ref, rec sec) to simplify the search of matching features (stereo matching) along the epipolar lines."

by deleting "to simplify the search of matching features (stereo matching) along the epipolar line"

and then by adding the following:

"Stereo-rectification allows to restrict the search of corresponding features from the entire image to a single line. For any point p in the reference view, the corresponding point p' in the secondary image, provided that it exists, lies on the epipolar line of p, that is EL p. Analogously, p lies on the epipolar line of p', EL p'. There is a correspondence between the epipolar lines of the two views for images taken with pinhole cameras. In this case the epipolar lines are said to be conjugate. The purpose of rectification is that of resampling the images in such a way that matching points are located on the same row (epipolar lines become horizontal), thus simplifying the search of matching features and allowing to use conventional stereo matching algorithms."

Comment on amt-2022-61 from Anonymous Referee #2

8. Line 279 / Fig. 8:

You describe that you conduct a stereo image rectification in order to make the stereo analysis easier, aren't you? In that case the y-component is usually zero (which you write in line 285). Is that correct? Also the disparity (parallax) should have values between 0 and infinity (or negative). But in Fig. disparities are both negative and positive. A negative disparity would mean that the observed point is behind the cameras. Or do I miss something?

Authors' reply:

Image rectification is indeed done to make the search of matching features easier. In this way matching cloud pixels are found on the same horizontal lines ($\Delta y$=vertical disparity=0). Vertical disparity is 0 as this is the main purpose of rectification. If this was not the case this would introduce an "epipolar" error that would affect the performance of the stereo matching. Once the images are rectified the only disparity that we are left with is the disparity along the x axis. This is measured, for each pair of matching features with image coordinates (h, k), in the rectified reference image, and (h+$\Delta x$, k), in the rectified secondary image, as X disparity= $\Delta x$=x(rect_sec)-x(rect_ref).

Following are two examples of negative and positive disparity, respectively.

Case 1: Negative disparity - Points closer to the cameras

In this case disparity is negative as for any given pair of matching features it turns out that

x rec sec < x rec ref (see x values for matching cloud structures).

[Figure]

Case 2: Positive disparity: Points farther away from cameras

In this case disparity is positive as for any given pair of matching features it turns out that

x rec sec > x rec ref (see x values for matching cloud structures).

[Figure]

For further clarity we can refer to the following drawing, which is an alternative way of looking at things, meant to show, from geometric principles, the fact that a negative disparity is actually expected for any given point $P_1$ at a closer distance to the reference camera than to the secondary one and vice versa (positive disparity), for any given point $P_2$ closer to the secondary camera than to the reference one.

[Figure]

$P_0$ is equidistant from the two cameras, $P_{0,l}$ and $P_{0,r}$ have both the same coordinates (h,k) in the images anddisparity along x is nul

$P_1$ is closer to the left camera (reference), $P_{1,l}$ coordinates in the reference image are still (h,k) and $P_{1,r}$ coordinates in the secondary image is (h+dx) with the disparity dx < 0

$P_2$ is further away from the left camera (reference), $P_{2,l}$ coordinates in the reference image are still (h,k) and $P_{2,r}$ coordinates in the secondary image is (h+dx) with the disparity dx > 0

For further clarity we redo figure 8 that now includes the x and y axis coordinates for the two zoomed image insets, which correspond to the red and blue rectangles drawn in the rectified images, containing the uppermost part of the cloud top. This should highlight the fact that, for any given pair of matching features, disparity is negative if rectified reference x > rectified secondary x. We also rotate the input images 90 degrees clockwise for consistency with the along track direction x and add axis grids in order to emphasize the fact that after rectification the y disparity becomes 0.

Moreover, we change color scale to improve on data visibility. For consistency we change color scale throughout the manuscript (see Fig. 11a, 12a and 12b).

Lines 289-293 (pp. 19) (lines 333-337, pp. 19 in the revised manuscript) we rephrase the following sentence:

"The disparity dx associated to each tie point is the distance between two corresponding points in the rectified images (see Fig. 8, step 3). Cloud pixels closer to the camera (i.e. cloud top) having larger difference in relative shift along x in the two rectified images are associated to larger disparity values (deep red points), whereas points farther away from the camera (associated to lower difference in relative shift) are associated to lower disparities (light blue/brown pixels)."

as follows:

"The disparity dx associated to each pair of matching features is the distance between two corresponding points in the rectified images (see Fig. 8, step 3, dx=rect sec x – rect ref x). Cloud pixels closer to the cameras (i.e. cloud top) are associated to negative disparity values (deep blue points) as, for any given pair of matching features, rect sec x < rect ref x, whereas for points farther away from the cameras, associated to positive disparity, rect sec x > rect ref x (orange/red pixels)."

We also add in the caption of figure 8:

"Notice that Ref and Sec have been rotated 90 degrees clockwise for consistency with the orientation of the along track direction x"

9. Fig. 10:
What explains the large differences in Fig. 10F for A10 and A11? Shouldn't the differences at least be symmetrical / similar to A1/A2? Similarly Fig. 10b. What could be a reason?
Authors' reply:
As we have already pointed out in reply to similar remarks from reviewer 1, with respect to Figure 10, we have repeated the calculations and accordingly updated the figure that now includes calculations for all scenarios.
In doing so, and in particular with respect to Fig. 10f, we notice that the mean Y error for configurations 1-3 now becomes 5 m/s whereas previously it was about 15 m/s and differences become symmetrical.
We then correct the sentence, lines 357, 358 (pp. 24) (lines 406, 407 – pp. 25 in the revised manuscript):
«The mean difference (its absolute value) along z is less than 25 m while it is less than about 5 m and 15 m along x and y, respectively.»
as follows:
«The mean difference (its absolute value) along z is less than 25 m while it is less than about 5 m along x and y.»

In the abstract we also correct, lines 9-11 (pp. 1) (lines 12-14 – pp. 1 in the revised manuscript):
"The accuracy of the retrieval of cloud topography is quantified in terms of RMSE and bias that are respectively, less than 25 m and 15 m for the horizontal components and less than 40 m and 25 m for the vertical component."
as follows:
"The accuracy of the retrieval of cloud topography is quantified in terms of RMSE and bias that are respectively, less than 25 m and 5 m for the horizontal components and less than 40 m and 25 m for the vertical component."

With respect to Figure 10b, concerning the skewed distribution of the error along z for the A9-A11 views, we add the following sentence, after (lines 357, 358 – pp. 24) (lines 406, 407 - pp. 25 in the

revised manuscript) "The mean difference (its absolute value) along z is less than 25 m while it is less than about 5 m along x and y.":

"The skewed distribution of the error in Figure 10b, for the views A9-A11, may be due to the fact that fewer cloud features are visible as the clouds are less illuminated by the sun, with larger portions of the cloud field shaded, as it can be seen from Figure 5g, 5h and 5i."

Comment on amt-2022-61 from Anonymous Referee #2

10. Sec. 5.3, comment: It is good that you mention possible differences due to the different distance estimation methods.

Comment on amt-2022-61 from Anonymous Referee #2

11. Fig. 12:

A sigma of 22.85 m/s in Fig. 12a for Vz seems a bit large considering the histogram.

Authors' reply:

Thanks for this remark. That is correct. By redoing the calculations we found out that about 2% of the total stereo retrieved points, those shown in red in the following figure,

[Figure]

are associated to unrealistically high values of Vz with abs(Vz)>20m/s. As it can be seen these points are mostly located over the cloud edges or in shaded regions where stereo reconstruction is expected to be less accurate. We therefore screen out these points and by redoing the calculations we now find a sigma of about 5.1 m/s , 1.1 m/s and 6.8 m/s for z, x and y, respectively and mean values of about 1.6 m/s, 6.4 m/s and 5.9 m/s for z, x and y, respectively.

For consistency with these changes, we redo Fig. 12a, extend the range of values, for the x axis of the Vz histogram, up to 20 m/s and also update (line 408-410, pp.28) (lines 461, 462 – pp. 30 in the revised manuscript) the following phrase:

"The horizontal velocity components show that the cloud is moving along the diagonal direction with a mean velocity of (6.5, 6.0) m/s."

by rewriting:

"The horizontal velocity components show that the cloud is moving along the diagonal direction with a mean velocity of (6.4, 5.9) m/s."

and the following phrase (line 410, pp. 28) (lines 463 – pp. 30 in the revised manuscript) as well:

"in the y direction the distribution is wider ($\sigma$ = 6.5 m/s)",

by rewriting:

"in the y direction the distribution is wider ($\sigma$ = 6.8 m/s)"

Moreover, before the following sentence (line 411,412, pp.28) (lines 466-468 – pp. 30 in the revised manuscript):

"With regard to the GE velocity derived from the GT envelopes, the mean velocities of 0.6 m/s, 6.5 m/s and 6.1 m/s are consistent with the ST mean velocities."

we add:

"It should be noticed that about 2% of the total number of stereo retrieved points, mostly located over the cloud edges or in shaded regions where stereo reconstruction is expected to be less accurate, are associated to unrealistically high values of Vz with abs(Vz)>20 m/s. Such points were filtered out."

Comment on amt-2022-61 from Anonymous Referee #2
12. Line 443:
"attitude" → "altitude"?
Authors' reply:
Line 443 (pp. 30) (line 501 – pp. 32 in the revised the manuscript), we mean indeed "attitude" in that by estimating a mean velocity over 100s we expect to smooth out the contribution to the error associated with satellite orientation.

---

## Author Comment (AC4)

Dear AMT Editors and Reviewers,

Thanks for starting the review process and for the precious comments and suggestions provided. In what follows we list the comments expressed by the handling associate editor (paragraphs written in blue color), provide our answers (paragraphs written in black color) to the raised points and explain how we addressed each point in the revised manuscript.

Comment by the handling associate editor :
Dear authors,
interesting work! I decided to go ahead and start the review and discussion. Nevertheless, I want to point out two things which you might consider in the revised version, once the reviews are available:
(1) concerning figure 2 it is mentioned that "most" of the artefacts were excluded from the analysis. Why only "most" and why did you not exclude them from figure 2?

Authors' reply :
In the initial version of the manuscript, figure 2 shows simulations A1, A4, A7. In the edited version of the manuscript, we now replace simulations A4 and A7 with A6 and A11, respectively, for consistency with figure 6 (that becomes figure 5 in the revised manuscript), showing the simulations for the cumulus case, and figure 1.

In this respect in section 2.1, lines 109-110 (pp. 4) (lines 126-127 – pp. 5 in the revised manuscript) we replace the following phrase :
« Figure 2 shows simulations of some of the successive CLOUD observations corresponding to acquisitions A1 (far from nadir), A4 and A7 (close to nadir). »
with the following one :
« Figure 2 shows simulations of some of the successive CLOUD observations corresponding to acquisitions A1 (far from nadir), A6 (close to nadir) and A11 (far form nadir and on the diametrically opposite view to A1 with respect to nadir). »

In this respect in section 3.1.3, lines 195-197 (pp. 12) (lines 280-283 – pp. 16 in the revised manuscript), we replace the following sentence:
« Figure 2 shows 3DMCPOL simulations for the acquisition A1 (T 0 ), off nadir (top three figures), acquisition A4 (T 0 + 60 s) (middle images - closer to nadir) and acquisition A7 (T 0 + 120 s) approximately at nadir (bottom figures), respectively. »
with the following one:
« Figure 2 shows 3DMCPOL simulations for the acquisition A1 (T 0 ), off nadir (top three figures), acquisition A6 (T 0 + 100 s) (middle images - approximately at nadir) and acquisition A11 (T 0 + 200 s) off nadir but on the diametrically opposite view to A1 with respect to nadir (bottom figures), respectively. »

The calculations presented in this paper, namely those shown in figure 8, 9(b), 11(a) and 12, were obtained using acquisitions A5 and A6, close-to-nadir views. With regard to such simulations, we use only the central part of the image as shown in the input images (see figure 8 - step 1 and figure 11a - step 1). Although some artifacts associated to the cyclic conditions used in the Monte Carlo code are visible, this non realistic effect does not affect the results obtained.

For more clarity, in section 2.1 lines 116, 117 (pp. 7) (lines 135-137 – pp. 7 in the revised manuscript) we replace the following sentence :
« Finally, there are image artifacts, mostly visible in the close-to-Nadir views (A4, A7), which are associated to the cyclic replication of the cloud field via the Monte Carlo code. »
with the following one :

« Finally, there are image artifacts, mostly towards the periphery of the images and especially visible in the close-to-Nadir view (A6) and the off-nadir acquisition (A11), which are associated to the cyclic replication of the cloud field via the Monte Carlo code. »

Still in the same section 2.1, lines 117-119 (pp. 7) (lines 137-139 – pp. 7 in the revised manuscript), we replace the following sentence :
« Most of these artifacts are excluded from the calculations presented in this work, specifically when performing stereo processing. This is achieved by opportunely setting the boundaries of the region of interest (ROI) of the images to a replicated domain. »
with the following phrase :
« The calculations presented in this paper, namely those corresponding to Fig. 8, 9(b), 11(a) and 12, were obtained using acquisitions A5 and A6. For such calculations, the region of interest (ROI) used for stereo processing corresponds to the central part of the images (see input images in Figure 8 - step 1 and Figure 11a - step 1). Although some artifacts, due to the cyclic conditions of the Monte-Carlo simulations, are visible, they do not affect in any way the results obtained. »

For more clarity, we also add in the caption of figure 8 the following sentence: « These calculations were obtained for acquisition A5. ».

Comment by the handling associate editor :
And (2) I would like to have some motivation why two completely different approaches were chosen (MesoNH + 3DMCPOL, SAM + Matsuba). I assume that the 3DMCPOL calculations are expensive, but in fact the should be cheaper for the cumulus case than for the deep convective case. The text wasn't very clear about how many scenes were calculated with 3DMCPOL and I would suggest some comments at a more prominent place, like the abstract and the summary/conclusions.

Authors' reply :
The realistic simulations presented in this paper have been obtained using the same principle, LES simulations coupled with a radiative transfer model. However, the tools used for simulating the two test cases presented are different. This is due to the time required for the development of the RT model and, in particular, the embedding of the geometric camera models in the 3DMCPOL code. Therefore, the first simulated test case is the one obtained with Mitsuba, readily available at the beginning of this works. If on the one hand Mitsuba allows to easily obtain a correct rendering of the images, on the other hand it relies on simplified optical properties (Henyey-Greenstein phase functions). Furthermore, the interface of 3DMCPOL with the geometric camera models allows to simulate observations for realistic orbits (calculated via Euclidium) and in a future version to take into account camera distortions. This will allow to test the algorithms under even more realistic conditions. For both cloud cases, 3DMCPOL calculations are highly time consuming. For this reason and due to time constraints, re-simulating the cumulus case was not possible. However, we think that presenting both test cases was worth it as it allows to show that the 3D reconstruction of the cloud envelope is not dependent from the type of cloud scene.

For more clarity, we swap section 3.1 with section 3.2 for consistency with the chronological order in which simulations were carried out.

Accordingly, we also modify in the abstract, in the introduction of section 3.2 and in the conclusions what specified as follows:

In the abstract lines 6-8 (pp. 1) (lines 6-8, pp. 1 in the revised manuscript), the following sentence:
« The latter are obtained via the radiative transfer model 3DMCPOL, for a deep convective cloud case generated via the atmospheric research model Meso-NH, and via the image renderer Mitsuba for a cumulus case generated via the atmospheric research model SAM. »

was replaced with:

«The latter are obtained via the image renderer Mitsuba, for a cumulus case generated via the atmospheric research model SAM, and via the radiative transfer model 3DMCPOL, coupled with the outputs of an orbit, attitude and camera simulator, for a deep convective cloud case generated via the atmospheric research model Meso-NH. »

In the introduction of section 3, for clarity, we add :
As no real data are available, in order to develop and test the cloud envelope and cloud development velocity retrievals, we simulate C3IEL observations for two test cases (11x3=33 images for each case). The first case is a cumulus case generated via the atmospheric research model SAM and the image renderer code Mitsuba. The second test case, a deep convective cloud, was simulated via the atmospheric research model Meso-NH and the radiative transfer model 3DMCPOL. The latter allows to exploit more realistic phase functions and was coupled with the outputs of an orbit, attitude and camera simulator. This second simulation is thus more realistic than the first one and in the future will allow to account for image distortion and satellite orientation error. However, time constraints associated to the high computational cost of the 3DMCPOL runs did not allow to re-simulate the cumulus cloud.

In the conclusion section we replace the following sentence, lines 448-450 (pp. 33) (lines 506-508, pp. 35 in the revised manuscript):
"These were obtained via Meso-NH and the radiative transfer code 3DMCPOL for a deep convective cloud case and via SAM and the image renderer Mitsuba for a shallow cumulus case."
with the following sentence:
"These were obtained via SAM and the image renderer Mitsuba for a shallow cumulus case and with Meso-NH and the 3D radiative transfer code 3DMCPOL, coupled with an orbit and geometric model simulator, for a deep convective cloud case."

at the end of the conclusion section, lines 469 (pp. 33) (lines 529 – pp. 35 in the revised manuscript), we also add:

"The cumulus case was not simulated via 3DMCPOL because of time constraints associated with the high computational cost of such calculations. However, in the future, to generalize our results, we plan to test our methods for other cloud types, scenes and solar geometries. This will be done by taking into account radiometric noise and image distortion as well as satellite orientation and position errors. This will allow to quantify the degradation of the results here obtained for "perfect simulations.".